# Symbiont location, host fitness, and possible coadaptation in a symbiosis between social amoebae and bacteria

Longfei Shu[1,2†], Debra A Brock[2†], Katherine S Geist[2], Jacob W Miller[3], David C Queller[2], Joan E Strassmann[2]*, Susanne DiSalvo[3]*

[1]Environmental Microbiomics Research Center and Guangdong Provincial Key Laboratory of Environmental Pollution Control and Remediation Technology, School of Environmental Science and Engineering, Sun Yat-sen University, Guangzhou, China; [2]Department of Biology, Washington University, St Louis, United States; [3]Department of Biological Sciences, Southern Illinois University, Edwardsville, United States

**Abstract** Recent symbioses, particularly facultative ones, are well suited for unravelling the evolutionary give and take between partners. Here we look at variation in natural isolates of the social amoeba *Dictyostelium discoideum* and their relationships with bacterial symbionts, *Burkholderia hayleyella* and *Burkholderia agricolaris*. Only about a third of field-collected amoebae carry a symbiont. We cured and cross-infected amoebae hosts with different symbiont association histories and then compared host responses to each symbiont type. Before curing, field-collected clones did not vary significantly in overall fitness, but infected hosts produced morphologically different multicellular structures. After curing and reinfecting, host fitness declined. However, natural *B. hayleyella* hosts suffered fewer fitness costs when reinfected with *B. hayleyella,* indicating that they have evolved mechanisms to tolerate their symbiont. Our work suggests that amoebae hosts have evolved mechanisms to tolerate specific acquired symbionts; exploring host-symbiont relationships that vary within species may provide further insights into disease dynamics.
DOI: https://doi.org/10.7554/eLife.42660.001

*For correspondence:
strassmann@wustl.edu (JES);
sdisalv@siue.edu (SDS)

†These authors contributed equally to this work

Competing interests: The authors declare that no competing interests exist.

## Introduction

Relationships are complicated because each party has evolved to maximize its own interests. Mutualisms arise when different parties can contribute unique abilities or resources to one another, under conditions where exploitation is controlled (*Bronstein, 2015*). Mutualisms where one or both parties are microbial fall under the general category of symbiosis. Symbioses are rife with potential conflict (*Dale and Moran, 2006*; *Estrela et al., 2016*; *Garcia and Gerardo, 2014*; *Moran, 2007*; *Oliver et al., 2005*). Despite this, symbiotic relationships are pervasive and persistent (*Douglas, 2008*; *Moran et al., 1993*; *Wernegreen, 2017*; *Werner et al., 2015*). The stability and ubiquity of these interactions implies that conflict can be managed or minimized by partners over multiple generations. However, stability does not imply stagnation. Recent studies suggest that examining extant symbiotic associations may not reveal the underlying mutualism to parasitism continuum that drives evolutionary change (*Moran, 2007*; *Queller and Strassmann, 2018*). Indeed, the view that symbiotic associations fluctuate along a mutualism-to-parasitism continuum is increasingly appreciated (*McFall-Ngai et al., 2013*).

Intracellular endosymbiosis involves a particularly intricate dance between partners. Intracellular endosymbionts must invade, survive, and replicate within host cells and move to new hosts. To exploit their host niche, mutualistic endosymbionts evolve specialized lifestyles that parallel those of

**eLife digest** A species can benefit or be hurt by other species. For example, honeybees and flowering plants help each other to flourish, while lions and gazelles behave in ways that decrease each other's populations. Understanding these relationships is important for controlling pests and diseases.

Sometimes it is easiest to study such interactions by looking at simple ones that happen on a small scale. Amoebas are common soil organisms that have the same basic organization as human cells. They are much larger and more complex than the bacteria that also live in the soil. How exactly the amoebas and bacteria interact in the soil is an important question, particularly as some of the bacteria can also live inside amoebas. Does this intimate relationship help or harm the amoeba?

Shu, Brock, Geist et al. studied the relationship between a widely studied species of social amoeba and two species of bacteria that can live inside it. Some of the amoebas naturally contained one of the bacteria species, and others were infected with the bacteria in experiments.

Throughout the entire life cycle of the amoebas, the bacteria lived inside them. During one part of the life cycle, amoebas form so-called fruiting bodies, which release spores that can develop into new amoebas. Shu et al. found that both types of bacteria alter the structure of the fruiting bodies in ways that reduce how well the spores disperse.

One of the bacteria species, called *Burkholderia hayleyella*, harmed the amoebas a lot. It caused most harm to amoebas that do not naturally host the bacteria. This indicates that the amoebas that do host this species may have evolved to avoid its worst effects.

The amoebas have many similarities to the white blood cells that clear bacteria from the human body. Certain bacteria can get inside white blood cells, causing diseases such as tuberculosis. Understanding how bacteria harm amoebas might be useful for understanding such diseases, and developing treatments for them. Though the bacteria Shu et al. studied are not toxic to humans, they are closely related to bacteria that are harmful. It is therefore possible that some bacteria that infect humans first evolve to infect amoebas.

DOI: https://doi.org/10.7554/eLife.42660.002

intracellular pathogens (*Casadevall, 2008*; *McCutcheon and Moran, 2011*; *Soto et al., 2009*). Despite initial exploitative strategies, some endosymbionts become beneficial or even obligate for host survival, a situation common for many insect nutritional endosymbioses (*Douglas, 2009*). Strict vertical transmission of endosymbionts typically promotes a more tranquil relationship as the evolutionary fates of the two parties become increasingly interdependent (*Ferdy and Godelle, 2005*; *Hosokawa et al., 2016*). Host dependency also often leads to the reduction in size of symbiont genomes as they become more streamlined for life within their host, something true for both beneficial and pathogenic endosymbionts (*Dale and Moran, 2006*). Horizontally acquired facultative symbionts are more likely to retain active conflicts. This has been experimentally demonstrated by manipulating the jellyfish symbiont *Symbiodinium microadriaticum* towards solely vertical or horizontal transmission modes, leading to the evolution of more mutualistic or parasitic lineages respectively (*Sachs and Wilcox, 2006*). Though horizontal transmission can favor symbionts that behave more parasitically, selection can favor hosts that employ more severe countermeasures to limit symbiont entry or growth (*Nyholm and McFall-Ngai, 2004*; *Ratzka et al., 2012*; *Reynolds and Rolff, 2008*). Indeed, host-driven control can be extreme, as demonstrated by the ability of *Paramecium bursaria* to manipulate its facultative nutritional symbiont (*Chlorella* sp.) in a relationship that provides no apparent benefit for the imprisoned symbiont (*Lowe et al., 2016*).

Symbiotic relationships may be positive or negative under different environmental conditions (*Leung and Poulin, 2008*; *Pérez-Brocal et al., 2011*). An interesting example of context dependency occurs in the *Acyrthosiphon pisum-Hamiltonella defensa* symbiosis (*Oliver et al., 2003*; *Oliver et al., 2005*). *H. defensa* infection confers host resistance to parasitoid attack, with resistance being greater for hosts co-infected with *Serratia symbiotica* (*Oliver et al., 2006*). However, co-infection comes at a high fecundity cost, such that in parasitoid free environments, host fitness is reduced compared to uninfected counterparts (*Oliver et al., 2006*). This interplay between reproductive strategy, context, and evolutionary history illustrates the complexity and fluidity of symbiosis.

The facultative endosymbiosis between the social amoeba *Dictyostelium discoideum* and *Burkholderia* bacteria provides a promising system for revealing insights into symbiosis dynamics (*Brock et al., 2011*; *DiSalvo et al., 2015*). *D. discoideum* is a soil dwelling amoeba with an interesting life cycle involving unicellular and multicellular stages. During the unicellular stage, amoebae consume bacteria by phagocytosis and divide. When prey are scarce, amoebae aggregate by tens of thousands to form multicellular slugs that move towards heat and light, seeking out a location to form fruiting bodies. These fruiting bodies consist of a stalk of sacrificial dead cells which support a globular sorus containing hardy spore cells (*Kessin, 2001*). When spores are dispersed, they germinate into vegetative amoebae and the cycle continues. Processes employed by specialized immune-like cells (sentinel cells) during slug migration and fruiting body formation remove any remaining bacteria, typically producing bacteria-free sori (*Brock et al., 2011*; *Chen et al., 2007*; *Cosson and Lima, 2014*).

However, some *D. discoideum* isolates harvested from the wild are infected with *Burkholderia* symbionts (*Brock et al., 2011*). Infection persists in the lab throughout the social cycle, where intracellular bacteria can be visualized within spores and sori (*DiSalvo et al., 2015*). Infection can be terminated by treating hosts with antibiotics and induced by exposing naïve hosts to *Burkholderia*, thereby allowing us to easily mix and match partners and study subsequent fitness consequences (*Brock et al., 2016a*; *DiSalvo et al., 2015*).

*D. discoideum* also has a meiotic sexual process, that occurs much less frequently than the asexual proliferation process of binary fission that amoebae go through every few hours (*Bloomfield et al., 2010*). The asexual binary fission process results in lineages that can be quite different, though all of the same species. Thus, a lineage that has acquired a bacterial endosymbiont can evolve to tolerate it independent of uninfected lineages at least for the thousands of generations that might pass before sexual recombination occurs. Comparing such lineages can illuminate how natural selection operates in early stages of symbiosis.

The *Dictyostelium-Burkholderia* association is particularly compelling for studying the parasitism to mutualism continuum in endosymbiosis because the fitness consequences of infection are context dependent. Under standard laboratory growth conditions, *Burkholderia* infection is detrimental for hosts because it decreases slug migration and spore production (*Brock et al., 2016a*; *DiSalvo et al., 2015*). However, infection can be beneficial. Infection induces secondary carriage of edible bacteria (*Burkholderia* is typically inedible), which can increase host fitness in food scarce conditions when the carried food bacteria reseed new environments with a food source (*Brock et al., 2011*; *DiSalvo et al., 2015*). For this reason, field-infected host amoebae are also called 'farmers' and uninfected host amoebae are called 'non-farmers.' Additionally, infected hosts are less harmed by ethidium bromide exposure, possibly mediated by bacterial degradation of the toxin (*Brock et al., 2016a*).

We have identified three *Burkholderia* species associated with *Dictyostelium: B. agricolaris, B. hayleyella,* and *B. bonniea* (*Brock et al., 2018*). Most of our work has been conducted with the first two species, which differentially impact host fitness. Here we probe the interaction between host and symbiont genotypes (with regards to their association history) with infection outcomes using a set of previously field-collected and characterized amoebae isolates and their associated *Burkholderia* symbionts (*Brock et al., 2011*; *DiSalvo et al., 2015*). We use standard laboratory conditions in which symbiont infection is shifted towards host detrimental outcomes. Since all three *Burkholderia* species can be cultured on Petri plates with standard media, none of them are entirely dependent on *Dictyostelium* for survival.

We find that *B. hayleyella* is most detrimental for *D. discoideum* hosts in general, but is most costly to those first exposed to it in the lab. We also document symbiont localization and morphological symptoms in hosts throughout development. Although both species of bacteria can be observed within phagocytic vacuoles, *B. hayleyella* infects more cells and damages fruiting body structures. These morphological aberrations are also less severe in native-*hayleyella* hosts, suggesting that native hosts have evolved mechanisms to withstand symbiont colonization.

# Results

## Impact of B. agricola and B. hayleyella on *D. discoideum* spore production

Our first goal was to clarify how infection of *Dictyostelium* by *Burkholderia* symbionts differentially influences host fitness. We used 12 natural isolates of *Dictyostelium*, each from three original conditions, uninfected with *Burkholderia,* infected with *B. agricolaris,* or infected with *B. hayleyella* (*Table 1*). We refer to these three types respectively as naïve hosts, native-agricolaris hosts, and native-hayleyella hosts. The word 'host' always means *D. discoideum,* and sometimes refers to potential hosts not actually infected with *Burkholderia.* Prior to each treatment, all host types were developed on nutrient media supplemented with bacterial food (*Klebsiella pneumoniae*). Amoebae, spores, cells, slugs, fruiting bodies, stalks, and sori refer only to *D. discoideum.*

We looked at *D. discoideum* spore viability and other measures under the following conditions: a) natural field state (naïve, native-*agricolaris*, native-*hayleyella*), b) those same hosts cured of *Burkholderia* (antibiotic treated), and c) condition after curing and reinfecting (with either *B. agicolaris* or *B. hayleyella*) (*Figure 1*). We quantified percent spore viability and number of spores produced (*Figure 2*, and *Supplementary files 1* and *2*). We multiplied these two measures to get a single main measure of fitness, viable spores produced.

## *D.discoideum* fitness does not differ by wild burkholderia infection status

We found that infection status in the field did not affect total viable spore counts for naïve, native-*agricolaris*, or native-*hayleyella* hosts (*Figure 2a* and *Supplementary file 1a*) (linear mixed model (LMM), $\Delta AIC = -1.57$, $\chi^2 = 5.57$, DF = 2, p = 0.062). Thus, field-infected native hosts do not seem to suffer any net fitness costs from infection by this measure.

## *D.discoideum* fitness does not decrease with antibiotic treatment

To make parallel comparisons when we newly infected *D. discoideum* with either of the two *Burkholderia* species, we first had to cure all hosts and be sure that curing in itself did not decrease fitness. We found that wild-collected hosts of our three categories did not experience lowered fitness after being cured with antibiotics (*Figure 2a and b*) (viable spore production: LMM, $\Delta AIC = 1.74$, $\chi^2 = 2.26$, DF = 2, p = 0.32, *Supplementary file 1b*). Antibiotic treatment actually increased viable

**Table 1.** *Dictyostelium discoideum* clones used for this study.

Clones are divided into specific sets each with naive, native-ag, and native-ha field-collected counterparts. They were collected from Virginia, North Carolina, and Texas as indicated.

| Set | Clone | Status | *Burkholderia* | Location collected | Date collected | GPS coordinates |
|---|---|---|---|---|---|---|
| 1 | QS9 | Naïve | None | Virginia-Mt Lake Biological Station | Oct. 15 2000 | N 37° 21′, W 80° 31′ |
| | QS70 | Native | *B. agricolaris* | Texas- Houston Arboretum | Jul. 15 2004 | N 29° 46′, W 95° 27′ |
| | QS11 | Native | *B. hayleyella* | Virginia-Mt Lake Biological Station | Oct. 15 2000 | N 37° 21′, W 80° 31′ |
| 2 | QS18 | Naïve | None | Virginia-Mt Lake Biological Station | Oct. 15 2000 | N 37° 21′, W 80° 31′ |
| | QS159 | Native | *B. agricolaris* | Virginia-Mt Lake Biological Station | May. 2008 | N 37° 21′, W 80° 31′ |
| | QS23 | Native | *B. hayleyella* | Virginia-Mt Lake Biological Station | Sep. 25 2000 | N 37° 21′, W 80° 31′ |
| 3 | QS17 | Naïve | None | Virginia-Mt Lake Biological Station | Oct. 15 2000 | N 37° 21′, W 80° 31′ |
| | QS161 | Native | *B. agricolaris* | Virginia-Mt Lake Biological Station | May. 2008 | N 37° 21′, W 80° 31′ |
| | QS22 | Native | *B. hayleyella* | Virginia-Mt Lake Biological Station | Sep. 25 2000 | N 37° 21′, W 80° 31′ |
| 4 | QS6 | Naïve | None | Virginia-Mt Lake Biological Station | Sep. 25 2000 | N 37° 21′, W 80° 31′ |
| | NC21 | Native | *B. agricolaris* | North Carolina-Little Butts Gap | Oct. 1988 | N 35° 46′, W 82° 20′ |
| | QS21 | Native | *B. hayleyella* | Virginia-Mt Lake Biological Station | Oct. 15 2000 | N 37° 21′, W 80° 31′ |

DOI: https://doi.org/10.7554/eLife.42660.003

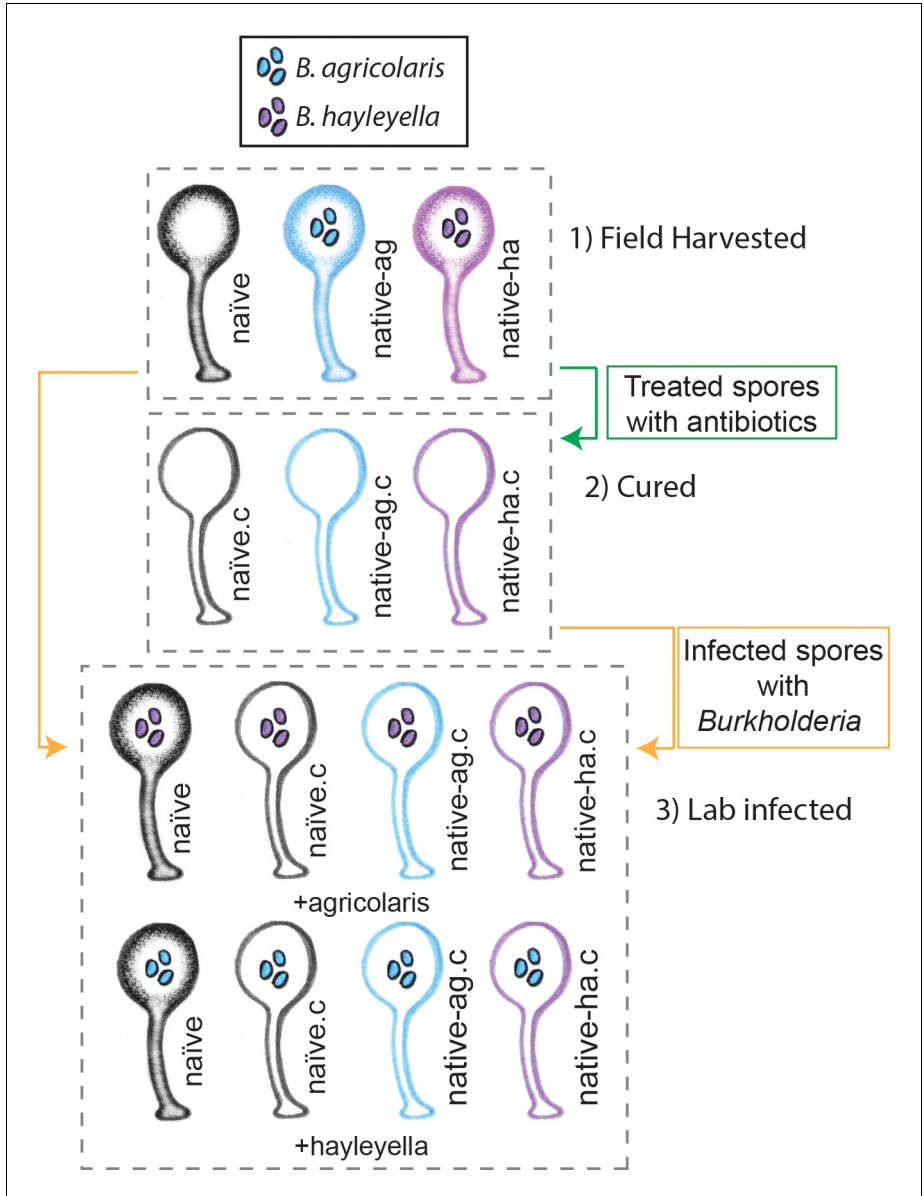

**Figure 1.** Illustration of host-symbiont pairs used throughout the study. *D. discoideum* clones were originally harvested from the wild in three different states: uninfected (indicated as naïve), or naturally infected with *B. agricolaris* or *B. hayleyalla* (indicated as native-ag, and native-ha respectively). Clones were treated with antibiotics to eliminate symbionts and are indicated with a '.c'. Clones were subsequently exposed to *Burkholderia* to initiate new infections. Thus, experimental types include 1) Field harvested, 2) cured, and 3) lab infected hosts.
DOI: https://doi.org/10.7554/eLife.42660.004

spore production of native-*agricolaris* hosts compared to uncured native-*agricolaris* hosts (LMM, $\Delta$AIC = −5.68, $\chi^2$ = 9.68, DF = 2, p = 0.008; *Figure 2a and b*).

### *D.discoideum* fitness decreases with exposure to burkholderia

To test the effects of *Burkholderia* on all types of field-collected hosts, we grouped host types into four sets, with each set representing a unique naïve, native-agricolaris, and native-hayleyella host (*Table 1*). We then infected each cured host type within the set with each *Burkholderia* from the native hosts within the same set. We compared viable spore production of cured *D. discoideum* hosts versus those same hosts artificially infected with *B. agricolaris* or *B. hayleyella* (*Figure 2b, c*

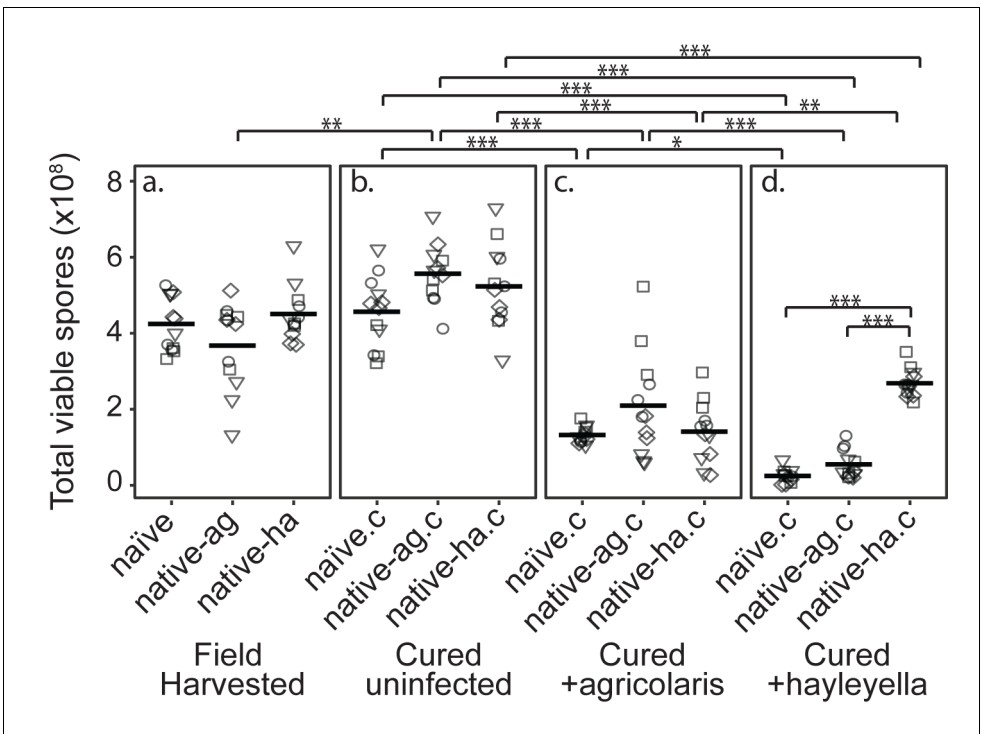

**Figure 2.** *Burkholderia* Infections Differentially Alter Spore Viability According to *Burkholderia* Species and Host Background. Total viable spores were determined for naïve and native hosts in their field harvested (**a**), cured (**b**), *B. agricolaris* lab-infected (**c**), and *B. hayleyella* lab-infected state (**d**). Four clones were measured for each type with three replicates for each (squares, triangles, circles, and diamonds represent set 1–4 clones respectively). Spore viability for wild harvested *B. agricolaris* and *B. hayleyella* host clones is higher than their cured-re-infected counterparts. Notably, spores from infected *B. agricolaris* and *B. hayleyella* native hosts (either naturally infected or cured and re-infected with their original *Burkholderia*) have a higher fitness than *Burkholderia* infected non-native counterparts. Bars represent significant differences (p < 0.05, and as indicated in supplemental tables).
DOI: https://doi.org/10.7554/eLife.42660.005
The following figure supplement is available for figure 2:

**Figure supplement 1.** Total Spore Number and Percent of Viable Spores for *Burkholderia* Infections in Diverse Host Backgrounds.
DOI: https://doi.org/10.7554/eLife.42660.006

*and d*). We found an overall effect on total spore viability with the addition of *Burkholderia* to all antibiotic-cured hosts (LMM, $\Delta$AIC = −198.81, $\chi^2$ = 210.81, DF = 6, $P \ll 0.001$). Addition of either *B. agricolaris* (Wald $t$ = −10.19, DF = 96, $P \ll 0.001$) or *B. hayleyella* ($t$ = −13.58, DF = 96, $P \ll 0.001$) to any of the cured hosts decreased their fitness (*Figure 2*; *Supplementary file 2*).

We then asked whether *D. discoideum* hosts are adapted to the *Burkholderia* species they carried in the field (using the same set strategy described above). Overall, the addition of *B. hayleyella* to *D. discoideum* led to significantly lower viable spore production than did the addition of *B. agricolaris* (Wald $t$ = −4.48, DF = 96, $P \ll 0.001$) (*Figure 2*). We also tested for an interaction between native host status and which *Burkholderia* species was added. There was an interaction effect on total viable spore production (LMM, $\Delta$AIC = −199.93, $\chi^2$ = 215.92, DF = 8, $P \ll 0.001$) (*Supplementary file 2e*). To address specific adaptation, we performed separate tests for each *Burkholderia* species added. When *B. hayleyella* was added to the three cured hosts, native-*hayleyella* hosts had higher fitness than did either native-*agricolaris* or naïve hosts (both $P \ll 0.001$, *Figure 2d*; *Supplementary file 2e*). In contrast, native-*agricolaris* hosts did not have significantly higher fitness with the addition of *B. agricolaris* than either native-*hayleyella* or naïve hosts (both p > 0.05, (*Figure 2c*; *Supplementary 2e*). However, there was a trend in the direction of native-*agricolaris* doing best (*Figure 2c*). These results indicate that native-*hayleyella* hosts are adapted to colonization by their field-acquired symbionts.

### *D.discoideum* morphology and burkholderia infected state

We next examined host morphology and symbiont localization at several stages of the *D. discoideum* life cycle. Using transmission electronic and confocal microscopy, we examined one *D. discoideum* clone for each host type outlined above and in *Figure 1* (QS9 for the naïve, QS70 for the native-*agricolaris*, and QS11 for the native-*hayleyella* host). These were either in an uninfected state or infected with one representative of *B. agricolaris* (Ba70 from QS70) or of *B. hayleyella* (Bh11 from QS11).

## Food bacteria location inside *D. discoideum* uninfected with burkholderia

D.*D. discoideum* morphology without *Burkholderia* in both naïve and cured native hosts, has vegetative cells that harbor no intracellular bacteria but contain many empty multilamellar bodies inside vacuoles (*Figure 3a* and *Figure 3–figure supplement 1*). Multilamellar bodies are predominantly composed of amoebae membranes and are thought to be a byproduct of bacterial digestion as they are produced when amoebae are fed bacterial food (*Denoncourt et al., 2016*; *Paquet et al., 2013*). Confocal microscopy of vegetative amoebae from this same host set grown with GFP-labeled food bacteria (*K. pneumoniae*) contain little to no intracellular GFP, suggesting that they have digested food bacteria by the time of fixation (*Figure 4a*).

After bacterial food has been depleted, vegetative cells aggregate to form multicellular migratory slugs. In accordance with our observations that all bacteria were killed and digested by vegetative cells, we found no intact bacteria in naïve or cured native host slugs (*Figure 5a*). In addition, slug cells were in general compacted with electron dense materials and contained no vacuoles or multi-lamellar bodies (*Figure 5a*).

Ultimately, slug cells differentiate into fruiting bodies consisting of dead stalk cells that support a sorus containing reproductive spore cells. In uninfected fruiting bodies, we did not detect bacteria in stalk cells or spores (*Figures 6a* and *7a*). Instead spores were packed with electron dense materials with no food vacuoles or multi-lamellar bodies (*Figure 6a*). Stalk cells showed plant cell-like characteristics, having a cellulosic cell wall and containing a single large vacuole (*Figure 6a*). Inside the large vacuole, there were some mitochondria and other cellular materials but no bacteria (*Figure 6a*). Taken together, these results suggest that the food bacterium we used, *Klebsiella pneumoniae,* was efficiently cleared during the social cycle from amoebae uninfected by *Burkholderia* and the amoebae then aggregate and produce bacteria-free fruiting bodies.

## Burkholderia location inside *D. discoideum*

When *D. discoideum* is infected with *Burkholderia,* we find it in amoebae, slug cells, spores, and stalk cells. Using confocal microscopy and RFP labeled *Burkholderia* strains, we were able to specifically identify high levels of *Burkholderia* inside host amoebae (*Figure 4b,c*). *B. hayleyella* was present in more of the amoebae than *B. agricolaris* was (*Figure 4b,c*). The higher number of *B. hayleyella* bacteria may account for the more detrimental fitness consequences it imposes (*Figure 2*).

The food bacterium *K. pneumoniae* labelled with GFP was occasionally observed in *Burkholderia* infected amoebae, particularly those infected by *B. agricolaris* (*Figures 4b* and *7b*). This suggests that *Burkholderia* colonization may partially impede food digestion, thereby allowing co-colonization of secondary bacteria and contributing to the proto-farming phenotype (*Brock et al., 2011*; *DiSalvo et al., 2015*). However, confocal images demonstrate that *Burkholderia* are much more abundant in amoebae than are *K. pneumoniae*, indicating that the majority of intact intracellular bacteria observed via transmission electron microscopy (TEM) are most likely *Burkholderia* cells (*Figures 3*, *5* and *6*).

TEM of vegetative amoebae shows intact bacteria which we infer to be *Burkholderia* surrounded by multi-lamellar bodies and located inside what appear to be food vacuoles (*Figure 3b,c*). Their undamaged appearance suggests that they are resistant to phagocytic digestion. In addition to the presence of intact intracellular bacteria, we also observe empty multi-lamellar bodies inside food vacuoles of infected amoebae. This suggests that digestion is not completely arrested during colonization, an unsurprising finding given that hosts continue to grow and multiply. Interestingly, we did not find multi-lamellar bodies containing intact bacteria secreted into the extracellular environment, indicating that *Burkholderia* may not be expelled from host cells via multi-lamellar body excretion.

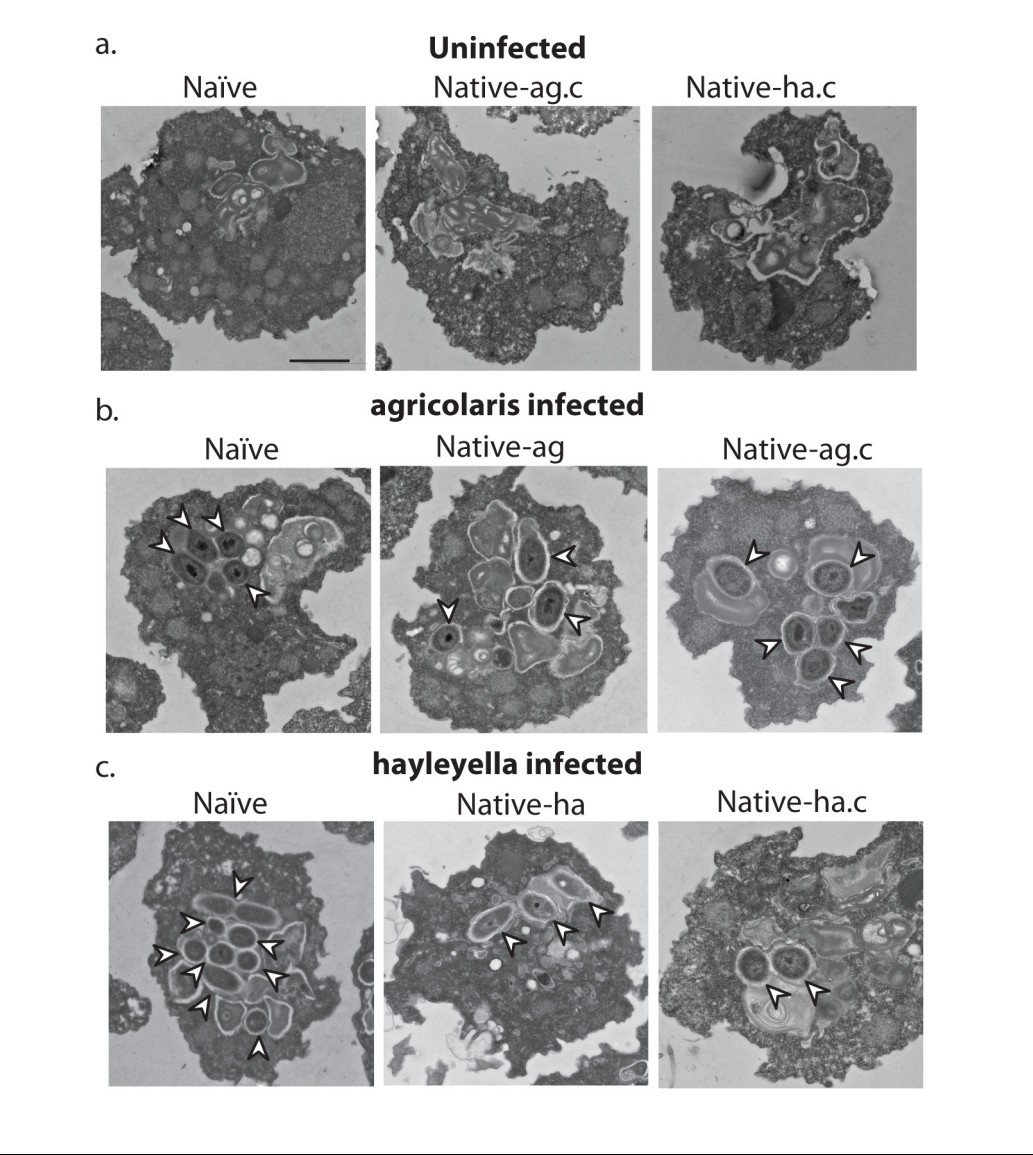

**Figure 3.** Bacterial cells are found within *Burkholderia* exposed vegetative amoebae. Transmission electron micrographs of vegetative amoebae show naïve and cured native amoebae with intracellular morphologies suggestive of active bacterial digestion with no evidence of intact intracellular bacteria (a). In contrast, bacterial cells can be found within *B. agricolaris* (b) and *B. hayleyella* (c) infected hosts. Arrows point to bacterial cells. More bacteria are observed in the *B. hayleyella* infected naïve host than in field harvested native-*hayleyella* and cured and re-infected native-*hayleyella* hosts (c). Bacterial cells appear to be within vacuole-like compartments. Scale bar (applicable to all): 2 um.
DOI: https://doi.org/10.7554/eLife.42660.007

The following figure supplement is available for figure 3:

**Figure supplement 1.** Multi-lamellar bodies excreted by vegetative amoebae.
DOI: https://doi.org/10.7554/eLife.42660.008

In *Burkholderia* infected hosts, intact bacteria are retained in food vacuoles throughout the transition to multicellular slugs, suggesting that bacteria stay within phagosomes throughout the aggregation stage (*Figure 5b,c*). Through TEM, we did not detect obvious morphological defects in infected slugs or differences between slugs infected with different *Burkholderia* species.

After fruiting body development, we find intracellular bacteria in both stalk and spore cells of *Burkholderia* infected hosts (*Figure 6b,c*). In infected stalk cells, intact bacteria reside in single large

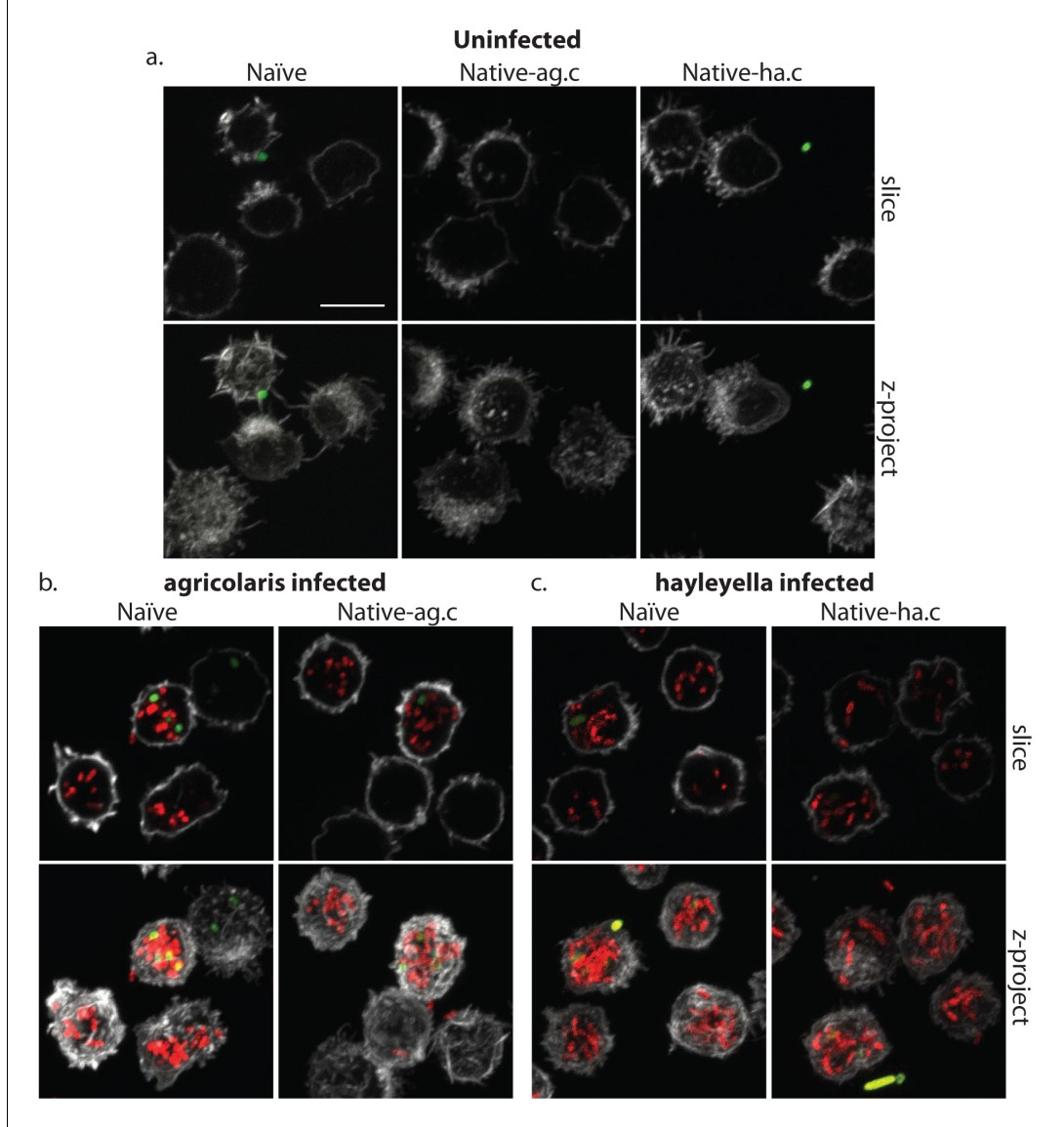

**Figure 4.** *Burkholderia* is found abundantly in colonized vegetative amoebae. Confocal imaging of fixed and stained vegetative amoebas show little to no intracellular bacteria in uninfected clones (a). However, abundant *Burkholderia* (*Burkholderia-RFP* shown in red) is found in *B. agricolaris* (b) and *B. hayleyella* (c) infected hosts. Occasional intracellular food bacteria (*Klebsiella*-GFP shown in green) is seen in *B. agricolaris* hosts (c). Spore coats are stained with phalloidin shown in grey. Scale bar 10 um.

DOI: https://doi.org/10.7554/eLife.42660.009

vacuoles inside the cellulosic cell wall (*Figure 6b,c*). In spore cells, bacteria remain within vacuoles (*Figure 6b,c*). We also observed bacterial cells outside spores but within the sorus, suggesting that bacteria may be carried interstitially into fruiting body structures, or they may be escaping pre-spore or spore cells within the sorus (*Figure 7b,c*).

We did not observe strikingly altered morphologies for *B. agricolaris* infected spore and stalk cells (*Figure 6b*). However, the spore and stalk cells of naïve hosts infected with *B. hayleyella* appeared to be morphologically aberrant (*Figure 6c*). We found numerous broken spores and signs that bacterial cells were escaping from damaged spores (*Figure 6c*). In addition, the whole stalk structure was often collapsed and filled with bacteria (*Figure 6c*). No clear cellulosic cell wall was observed in stalk cells, suggesting that *B. hayleyella* colonization inhibits the normal development of stalk cells or results in their disruption in naïve hosts and native-*agricolaris* hosts.

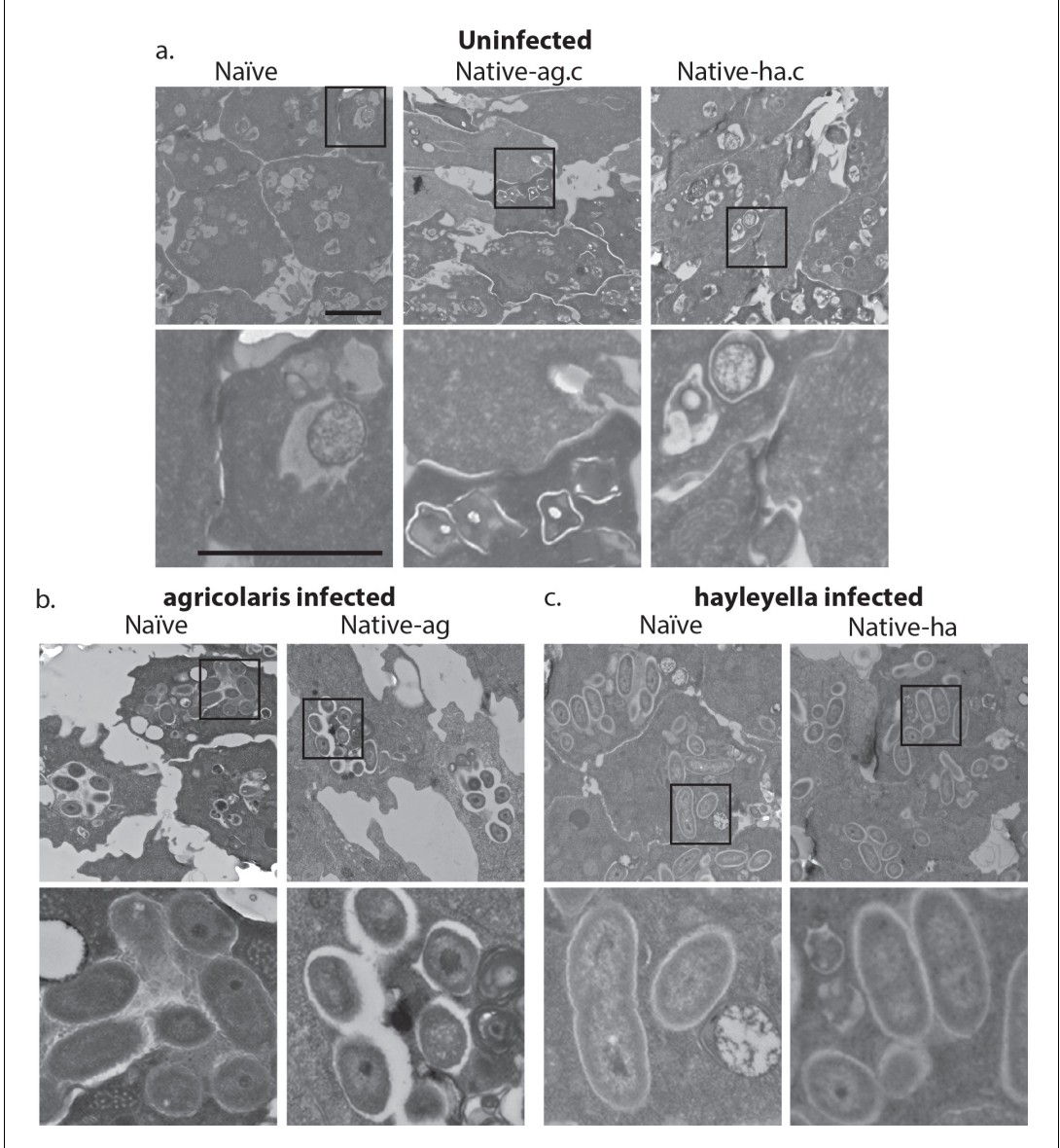

**Figure 5.** Intracellular bacteria are retained in naïve migrating slugs exposed to *Burkholderia* and in native *Burkholderia* hosts. Transmission electron micrographs of uninfected (a) show closely packed amoebae with internal structures reminiscent of previous bacterial digestion but without evidence of intact internal bacteria. In contrast, *B. agricolaris* (b) and *B. hayleyella* (c) infected slugs retain intracellular bacteria. Bottom panels represent magnified versions (see box) of upper panels. Scale bar (applicable to all panels in row) 2 um.

DOI: https://doi.org/10.7554/eLife.42660.010

In line with the visualized differences between the abundance of the two species of *Burkholderia* in amoebae, we find significantly more spores infected with RFP labeled *B. hayleyella* (mean = 88.8%) than similarly labeled *B. agricolaris* (mean = 35.3%) ($F_{2,4}$ = 191.33, p < 0.001) (*Figure 7d*). These results indicate that the degree of fitness detriment imposed by *B. hayleyella* could be a result of bacterial infection prevalence within the population. However, *B. agricolaris* displays a significantly higher intracellular density (mean = 10.55) within individual infected spores than *B. hayleyella* (mean = 7.45) ($F_{1,177}$ = 23.845, p < 0.001), while host background does not play a significant role in *Burkholderia* intracellular density ($F_{2,177}$ = 2.723, p = 0.068) (*Figure 7e*).

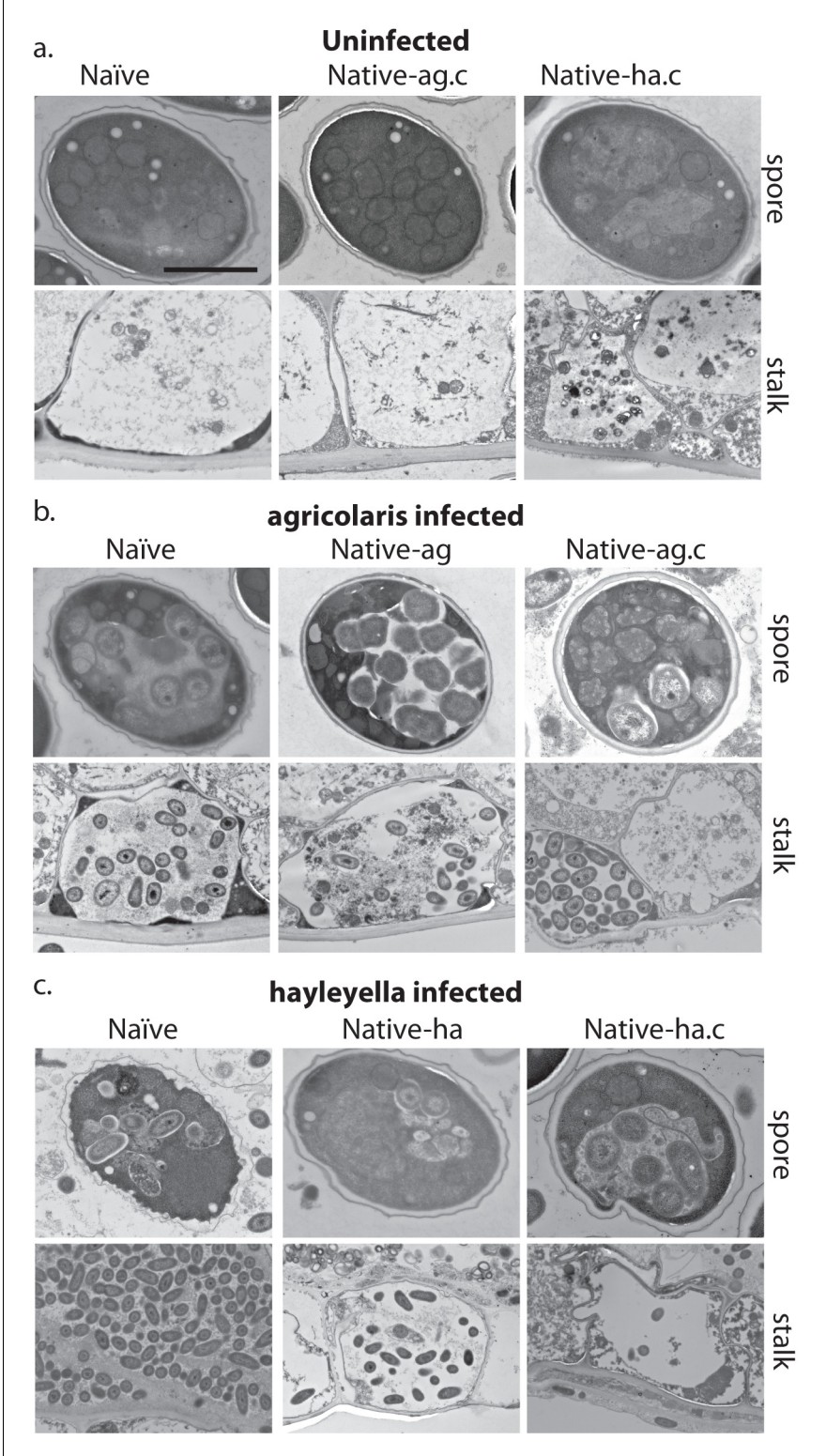

**Figure 6.** Bacterial cells are retained in spore and stalk cells from *Burkholderia*-exposed hosts. As visualized through transmission electron microscopy, (**a**) uninfected hosts form sturdy spores and stalk cells with no detectable bacteria. Spores and stalk cells retain intracellular bacteria in B. *agricolaris* (**b**) and B. *hayleyella* (**c**) hosts. Naïve *B. agricolaris* hosts appear structurally similar to uninfected cells while naïve *B. hayleyella* hosts have compromised spore coats and collapsed stalk structures filled with bacteria. Scale bar: 2 um.
DOI: https://doi.org/10.7554/eLife.42660.011

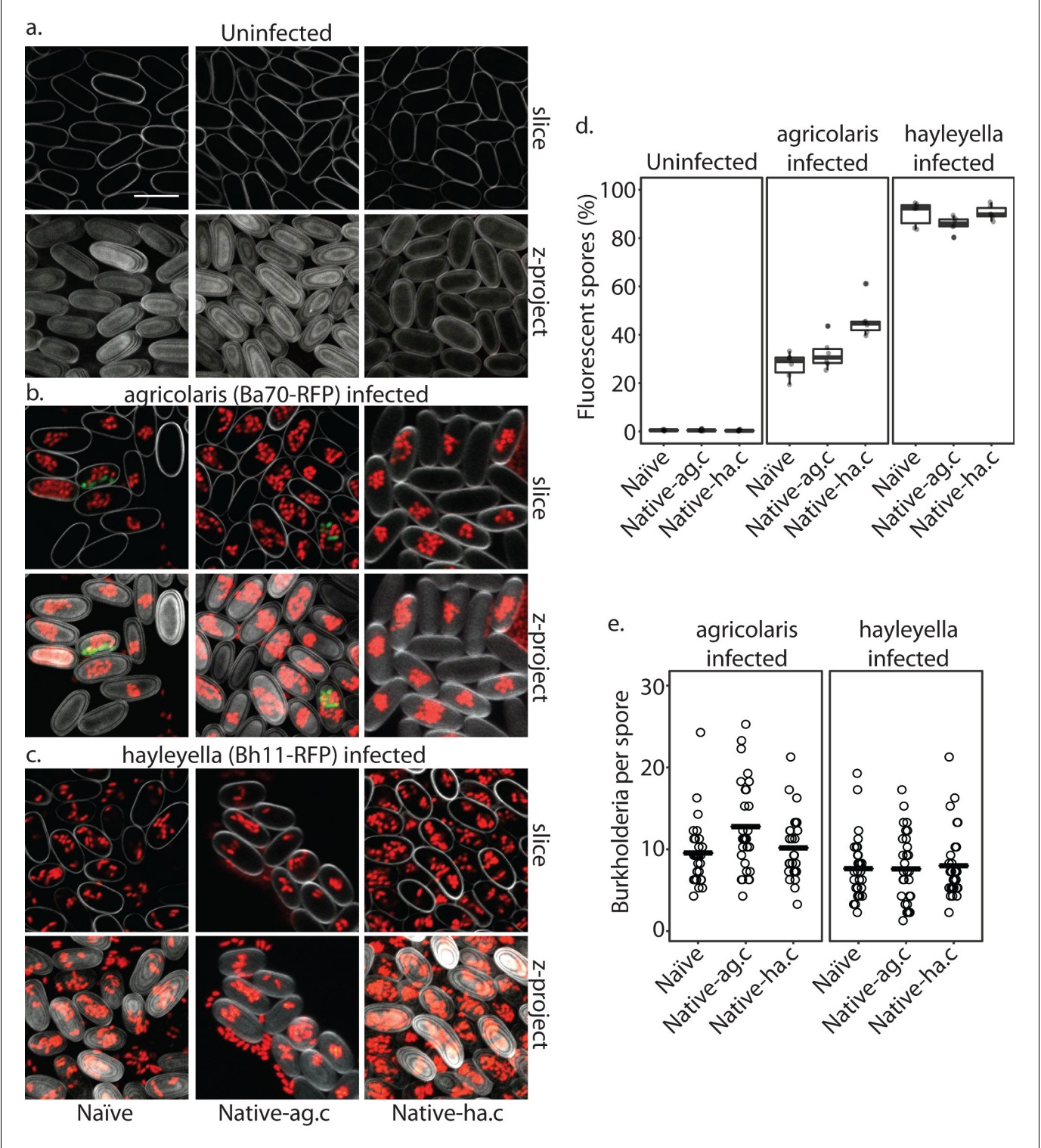

**Figure 7.** *Burkholderia* is retained in the sori of developed *D. discoideum* hosts and the percent of *Burkholderia* positive spores differs according to *Burkholderia* species. Confocal images show no intra- or extracellular bacteria in uninfected spores (a) Abundant *Burkholderia* is seen in *B. agricolaris* (b) and *B. hayleyella* (c) hosts, with more infected spores seen for *B. hayleyella* (d) but more *Burkholderia*-RFP cells detected per infected spore for *B. agricolaris* hosts (e). Co-infection by food bacteria is occasionally observed in *B. agricolaris* infected spores (b). For a-c: *Klebsiella*-GFP shown in green, *Burkholderia*-RFP shown in red, and calcofluor stain shown in grey. Top panels are image slices; bottom panels are max intensity projections of z stacks. Scale bar: 10 um.

DOI: https://doi.org/10.7554/eLife.42660.012

## Burkholderia impact on fruiting body morphology

Since Burkholderia are found inside D. discoideum, it is no surprise they impact the morphology of fruiting bodies. In their field-collected state, the three clones carrying no Burkholderia, or B. agricolaris or B. hayleyella differed in both stalk height and stalk volume ($F_{2,27}$ = 42.6, P ≪ 0.001, $F_{2,27}$ = 50.8, P ≪ 0.001, *Figure 8b*, *Supplementary file 3a*). The main pattern is that both height and volume were significantly lower in native-hayleyella hosts (*Figure 8*; *Supplementary file 3a*).

Native-*agricolaris* fruiting bodies were generally similar to the naïve host but taller than the native-*hayleyella* host (p < 0.001). In the native-*agricolaris* host, the spore masses often slid down their stalks or the fruiting bodies fell over, though the stalks were not significantly taller than in the

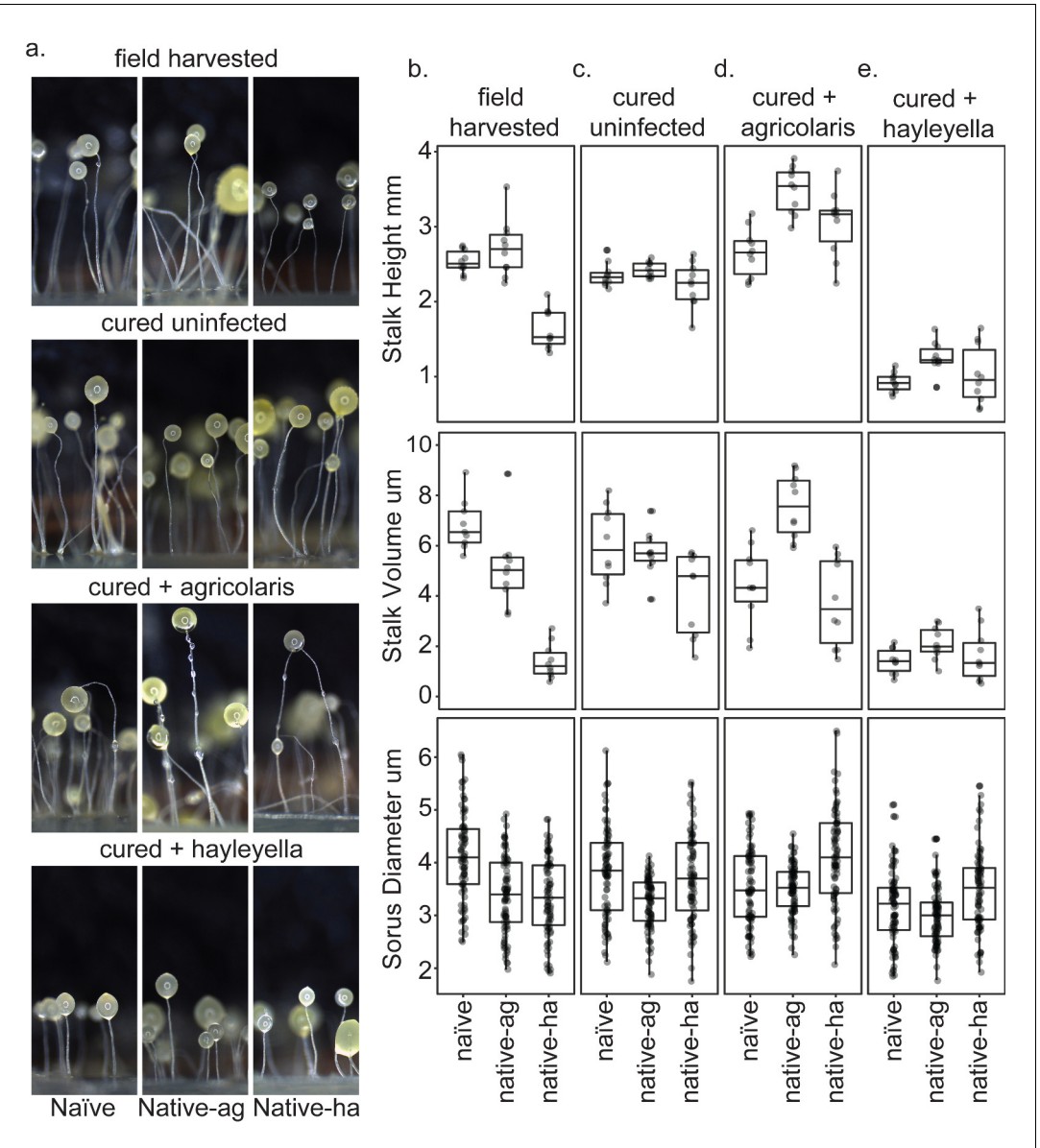

**Figure 8.** Fruiting body morphology is differentially altered by *Burkholderia* colonization. Macro photographs of fruiting bodies (a) show slightly different morphologies according to *Burkholderia* infection status. Sori measurements demonstrate that field collected native-*hayleyella* hosts produce shorter stalks and less voluminous sori (b). Cured hosts produce similar fruiting body measurements across host background (c). Cured hosts subsequently infected with *B. agricolaris* produce slightly taller stalks, which is most noticeable in cured and re-infected native-*agricolaris* hosts (d). Cured hosts subsequently infected with *B. hayleyella* all produce significantly shorter stalks with overall smaller fruiting body dimensions (e).
DOI: https://doi.org/10.7554/eLife.42660.013

naïve host. If the spores fall off their stalks, they will not have the advantage of facilitated transport by a vector that they would have at the top of the stalk (*Smith et al., 2014*).

There was also an overall difference in sorus diameter and sorus volume among the three hosts in their field state (*Figure 8*; $F_{2,236}$ = 25.4, $P \ll 0.001$, $F_{2,236}$ = 22.9, $P \ll 0.001$, *Supplementary file 4a*). Compared to the naïve hosts, both the native-*agricolaris* and native-*hayleyella* hosts had smaller sorus sizes (both p < 0.001) but were not different from each other.

Curing with antibiotics caused no significant change in any stalk or spore measurements (*Figure 8a,b,c*; *Supplementary file 3b*, *Supplementary file 4b*). However, when *Burkholderia* bacteria were added to the cured hosts, we saw species-specific effects on morphology (*Figure 8*; *Supplementary file 3c-d*, *Supplementary file 4c-d*). Overall, the addition of *B. agricolaris* changed stalk height ($F_{1,58}$ = 54.9, $P \ll 0.001$) significantly increasing it in two of the hosts (*Supplementary file 3c*). Stalk volume was not affected. Addition of *B. agricolaris* also significantly changed both sorus diameter and sorus volume ($F_{2,240}$ = 25.4, $P \ll 0.001$, $F_{2,240}$ = 21.2, $P \ll 0.001$). Native-*hayleyella* hosts had larger sori when infected with *B. agricolaris* (*Figure 8*; *Supplementary file 4c*).

The addition of *B. hayleyella* decreased both stalk height (*Figure 8*; $F_{1,58}$ = 366.4, $P \ll 0.001$) and stalk volume ($F_{1,58}$ = 120.2, $P \ll 0.001$, *Supplementary file 3d*) significantly in all three host types. Addition of *B. hayleyella* also affected sorus diameter ($F_{2,240}$ = 10.1, $P \ll 0.001$) and volume ($F_{2,240}$ = 11.3, $P \ll 0.001$) with a significant specific effect of smaller sori in the naïve host (*Figure 8*; *Supplementary file 4d*).

There were interaction effects between the amoebae hosts' native infection status and the species of *Burkholderia* added for all traits: stalk height ($F_{4,81}$ = 4.9, p = 0.0015), stalk volume ($F_{4,81}$ = 7.4, $P \ll 0.001$; *Figure 8*; *Supplementary file 3e*), sorus diameter ($F_{4,719}$ = 5.3, p = 0.0004) and sorus volume ($F_{4,719}$ = 4.5, p = 0.0014; *Figure 8*; *Supplementary file 4e*). However, these interactions were matters of degree of change and did not involve sign changes: for all four measurements, fruiting bodies with *B. agricolaris* were taller than those with *B. hayleyella* (*Figure 8*; *Supplementary file 3e*, *Supplementary file 4e*).

## Discussion

Here, we characterized *D. discoideum* infection by two symbiotic *Burkholderia* species, *B. hayleyella* and *B. agricolaris* (*Brock et al., 2018*). We looked at their impact on *D. discoideum* by comparing wild type, cured, and re-infected hosts. We assessed fitness measured as production of viable spores, and also evaluated morphological changes in amoebae, slugs, and fruiting bodies with numerical and microscopic data. We found that both *Burkholderia* species are a burden to *D. discoideum* under our experimental conditions. However, wild collected hosts did not differ in viable spore production according to whether or not they carried either species of *Burkholderia*. Even so, *D. discoideum* with their field-collected state of infection did differ in fruiting body dimensions, with uninfected hosts generally having taller, larger stalks, and larger sori. What explains the differences from experimental infection is unclear. Infection in the wild may be at a lower level than we used experimentally or may have initiated at a lower level that slowly amplified over time, allowing host acclimation to the metabolic costs of infection.

Once *D. discoideum* hosts are cured with antibiotics, so all comparisons can start from the same baseline, we found that there were few within treatment differences according to host type. The only exception to this is that native-*hayleyella* hosts produced more viable spores than did naïve or native-*agricolaris* hosts. This fitness difference is an indication of co-adaptation.

When we map naïve, native-agricolaris, and native-hayleyella host types from *Haselkorn et al. (2018)* on a previously generated phylogeny, we see no phylogenetic differentiation whatsoever between host types (*Douglas et al., 2011*). It is most likely that diverse amoebae isolates get infected with *Burkholderia* in nature, and then adaptations that provide these hosts with increased resilience are selected for over time. Perhaps the general lack of difference among host types with infection is due to sex, which not only recombines genes, but also exposes new clones horizontally to endosymbionts like *Burkholderia*. *Dictyostelium* recombination rates are high in natural populations (*Flowers et al., 2010*). The differences reported here among clones of *D. discoideum* indicate that specific co-adaptation in clones infected with *B. hayleyella* can occur during the long periods between sexual reproduction. Thus, reproduction by binary fission and vertical transmission of *B.*

*hayleyella* is likely to occur much more often than sexual reproduction which could result in horizontal transmission.

Our previous work demonstrated that *Burkholderia* infections have contextually dependent costs and benefits for their hosts. In food abundant conditions (which we used here) *Burkholderia* infections are generally detrimental to host fitness (*Brock et al., 2011*). However, when dispersed to food scarce conditions, *Burkholderia* infected hosts are able to transport food bacteria with them which restocks their food source and results in higher host fitness ('farming'). These fitness outcomes may result from the ongoing dynamics underlying long-term symbiosis.

A compelling question is what mediates this tolerance to *B. hayleyella* infection in its native host? It is possible that native hosts better inhibit infection events or better control intracellular replication of symbiont cells. The percent of spore cells infected in the population post symbiont exposure are not significantly different between a native-*hayleyella* host and naïve hosts, which does not support the idea that infection events are inhibited by hosts conditioned to that species. Further, we did not see a significantly different intracellular density of *B. hayleyella*-RFP across hosts from confocal micrographs, suggesting that intact spores from different hosts can accommodate similar numbers of bacterial cells. However, our TEM analysis qualitatively points to the idea that after infection, intracellular replication rates may differ between naïve and native-*hayleyella* hosts. TEM images of native and naïve *B. hayleyella* hosts consistently suggest a higher load of intracellular bacterial cells at each stage. For instance, naïve *B. hayleyella* hosts produce paltry stalks that appear to be overwhelmed by bacterial cells and infected spore cells that look on the verge of deteriorating. Neither of these extreme states were observed in the native host. This could be mediated by host countermeasures that control intracellular symbiont growth or disarm potential toxic symbiont byproducts.

Compared to *B. hayleyella*, *B. agricolaris* infections result in more modest (and statistically insignificant) drops in *D. discoideum* viable spore production for naïve and cured re-infected native-*agricolaris* hosts. *B. agricolaris* appears to be less invasive to the host population as it is found in only about a quarter of spore population after exposure. Despite this, it is maintained in infected populations throughout the social stage and over multiple social cycles (*DiSalvo et al., 2015*). However, *B. agricolaris* establishes a higher intracellular density within infected spores than *B. hayleyella*. These observations may be explained by reduced intracellular entrance efficiency coupled with higher intracellular replication rates. Alternatively, *B. agricolaris* could be more efficiently cleared by host mechanisms and requires a higher multiplicity of infection to overcome clearance, something that might help explain its more modest detrimental effect on host populations.

Earlier studies have identified other differences between naïve *D. discoideum* and those carrying *B. hayleyella* (*Brock et al., 2013*; *Brock et al., 2016b*; *Stallforth et al., 2013*). *D. discoideum* hosts carrying *B. hayleyella* harmed symbiont-free *D. discoideum* clones, causing them to lose in social competition (*Brock et al., 2013*). In another study we compared cured and uncured clones of *B. hayleyella* and found that slugs from uncured clones move less far across a Petri plate, a cost of infection (*Bates et al., 2015*). Interestingly, in the current study we detected no visible differences between cured and uncured slug cells. The recent discovery of sentinel cells as innate immune cells (*Chen et al., 2007*) made us wonder about how they fare with *Burkholderia* infected hosts. We found that *D. discoideum* hosts with *Burkholderia* do not produce as many sentinel cells but even so seem as resistant to toxins as uninfected lines with normal levels of sentinel cells (*Brock et al., 2013*; *Brock et al., 2016b*; *Stallforth et al., 2013*).

Another interesting recent result on the interaction between *D. discoideum* and bacteria involves the role of the lectin discoidin 1 (*Dinh et al., 2018*). Clones infected with *Burkholderia* produced much greater quantities of lectins early in the social stage compared to uninfected clones. These lectins coated the food bacterium *K. pneumoniae* allowing it to avoid digestion. This is undoubtedly just a beginning, though a fascinating one, in our understanding of how *Burkholderia* take over *D. discoideum* cellular machinery to change relationships with bacteria.

D.*D. discoideum* is already a popular system for examining the molecular mechanisms of bacterial pathogenesis for a variety of important pathogens (*Cosson and Soldati, 2008*). However, this work demonstrates several properties of this interaction that are distinct from other *Dictyostelium*-bacterial associations, possibly owing to its natural prevalence. For instance, *Bordetella bronchiseptica* can intracellularly infect vegetative *D. discoideum* amoebae and persist in sorus contents. However, in contrast to *Burkholderia*, *B. bronchiseptica* is localized extracellularly in sori rather than inside spore cells (*Taylor-Mulneix et al., 2017*). Infections of *D. discoideum* with other intracellular pathogens

such as *Legionella pneumophila* often produce secreted multilamellar bodies which harbor the bacterial pathogen (*Denoncourt et al., 2014*; *Paquet and Charette, 2016*), while in this study we found no evidence of *Burkholderia* excreted in multilamellar bodies. In addition, many bacteria resistant to amoebae are found packaged in multilamellar bodies, a process speculated to enhance their resistance to environmental stress (*Denoncourt et al., 2014*). Thus, *Burkholderia* symbionts are most likely employing alternative tactics to not only evade digestion, but to also evade expulsion.

*Burkholderia* is important in another model symbiosis system, that with the bean bug, *Riptortus pedestris* (*Takeshita and Kikuchi, 2017*). These bugs acquire *Burkholderia insecticola* horizontally. They reside in the bean bug gut where they are presumably active in nutrition. They are a good model because it is a facultative symbiosis and both partners can be cultured independently.

We are in a new age of symbiosis studies where we can apply Koch's principles of curing, reinfecting, and looking for evidence of disease. We can use genomics, experimental evolution and many other methods to ever more systems. The *D. discoideum-Burkholderia* system holds unique potential for studying eukaryote-bacterial associations. Given its natural occurrence, we can perform long-term ecological surveys, easily isolate new host-symbiont pairs, investigate naturally derived vs newly induced associations using a variety of partner pairing, and we can do experiments in evolution allowing each partner to evolve together or separately. In time, *B. hayleyella* and *B. agricolaris* in *D. discoideum* may be added to the classic symbioses of squid-vibrio, aphid-*Buchnera*, tsetse fly-*Wiggelsworthia*, legume-*Rhizobia* and more (*Bennett and Moran, 2015*; *Bing et al., 2017*; *Koehler et al., 2018*; *Werner et al., 2015*).

# Materials and methods

### Key resources table

| Reagent type (species) or resource | Designation | Source or reference | Identifiers | Additional information |
|---|---|---|---|---|
| Strain, strain background (*Dictyostelium discoideum*) | QS6 | *Douglas et al., 2011*, *Brock et al., 2011* | | Virginia-Mt Lake Biological Station |
| Strain, strain background (*D.discoideum*) | QS9 | *Douglas et al., 2011*, *Brock et al., 2011* | | Virginia-Mt Lake Biological Station |
| Strain, strain background (*D. discoideum*) | QS17 | *Douglas et al., 2011*, *Brock et al., 2011* | | Virginia-Mt Lake Biological Station |
| Strain, strain background (*D. discoideum*) | QS18 | *Douglas et al., 2011*, *Brock et al., 2011* | | Virginia-Mt Lake Biological Station |
| Strain, strain background (*D. discoideum*) | QS11 | *Douglas et al., 2011*, *Brock et al., 2011* | | Virginia-Mt Lake Biological Station |
| Strain, strain background (*D. discoideum*) | QS21 | *Douglas et al., 2011*, *Brock et al., 2011* | | Virginia-Mt Lake Biological Station |
| Strain, strain background (*D. discoideum*) | QS22 | *Douglas et al., 2011*, *Brock et al., 2011* | | Virginia-Mt Lake Biological Station |
| Strain, strain background (*D. discoideum*) | QS23 | *Douglas et al., 2011*, *Brock et al., 2011* | | Virginia-Mt Lake Biological Station |
| Strain, strain background (*D. discoideum*) | QS70 | *Douglas et al., 2011* | | Texas- Houston Arboretum |
| Strain, strain background (*D. discoideum*) | QS159 | *Brock et al., 2011* | | Virginia-Mt Lake Biological Station |

*Continued on next page*

*Continued*

| Reagent type (species) or resource | Designation | Source or reference | Identifiers | Additional information |
|---|---|---|---|---|
| Strain, strain background (*D. discoideum*) | QS161 | *Brock et al., 2011* | | Virginia-Mt Lake Biological Station |
| Strain, strain background (*D. discoideum*) | NC21 | *Francis and Eisenberg, 1993* | | NC-Little Butts Gap |
| Strain, strain background (*Burkholderia hayleyella*) | BhQS11 | *Haselkorn et al., 2018* | | isolated from QS11 |
| Strain, strain background (*B. hayleyella*) | BhQS21 | *Haselkorn et al., 2018* | | isolated from QS21 |
| Strain, strain background (*B. hayleyella*) | BhQS22 | *Haselkorn et al., 2018* | | isolated from QS22 |
| Strain, strain background (*B. hayleyella*) | BhQS23 | *Haselkorn et al., 2018* | | isolated from QS23 |
| Strain, strain background (*Burkholderia agricolaris*) | BaQS70 | *Haselkorn et al., 2018* | | isolated from QS70 |
| Strain, strain background (*B. agricolaris*) | BaQS159 | *Haselkorn et al., 2018* | | isolated from QS159 |
| Strain, strain background (*B. agricolaris*) | BaQS161 | *Haselkorn et al., 2018* | | isolated from QS161 |
| Strain, strain background (*B. agricolaris*) | BaNC21 | *Haselkorn et al., 2018* | | isolated from NC21 |
| Strain, strain background (*B. agricolaris*) | BaQS70-RFP.1 | *DiSalvo et al., 2015* | | modified from BaQS70 |
| Strain, strain background (*B. hayleyella*) | BhQS11-RFP.2 | This paper | | modified from BaQS11 |
| Strain, strain background (*Klebsiella pneumoniae*) | KpQS | Dictybase (http://dicty base.org/) | | |
| Strain, strain background (*K. pneumoniae*) | KpQS-GFP.1 | This paper | | |
| Recombinant DNA reagent | pmini-Tn7-KS-GFP | *Teal et al., 2006* | | |
| Recombinant DNA reagent | pmini-Tn7-gat-P1-RFP | *Su et al., 2014* | | |

## *D. discoideum* strains and culture conditions

We collected *D. discoideum* isolates from the field that were uninfected, or infected with either *B. agricolaris*, or *B. hayleyella*. *Table 1* describes host clone sets, location collected, and infection status. We used host sets 1–4 for the spore fitness assays and set one for all other experiments. We used *Klebsiella pneumoniae* obtained from the Dicty Stock Center (http://dictybase.org/StockCenter/StockCenter.html) as our food bacterium for *D. discoideum*. We grew all *D. discoideum* from

spores on SM/5 agar plates (2 g glucose, 2 g BactoPeptone (Oxoid), 2 g yeast extract (Oxoid), 0.2 g MgCl$_2$, 1.9 g KH$_2$PO4, 1 g K$_2$HPO4 and 15.5 g agar per liter) supplemented with *K. pneumoniae* at room temperature (21°C).

## Symbiotic bacterial strains

We used *D. discoideum*-associated *Burkholderia* previously isolated and sequenced to verify closest 16S identity (*Brock et al., 2011*; *DiSalvo et al., 2015*). *B. agricolaris* and *B. hayleyella* strains were isolated from QS70, QS159, QS161, and NC21, and QS11, QS23, QS22, and QS21 *D. discoideum* hosts respectively. According to multi-locus sequence typing all *B. hayleyella* strains belong to the same haplotype, *B. agricolaris* from QS70 and QS161 are the same haplotype and *B. agricolaris* from NC21 and QS159 are unique haplotypes (*Haselkorn et al., 2018*).

## Removal of symbiont from native *D. discoideum* hosts

We generated symbiont-free native host clones by tetracycline, or by ampicillin-streptomycin, treatment as previously described (*Brock et al., 2011*; *DiSalvo et al., 2015*). We confirmed loss of infection status using the spot test assay and PCR analysis of *Burkholderia*, and *K. pneumoniae* in *D. discoideum* sori as previously described.

## Lab infections

We collected stationary phase bacteria in starvation buffer (2.2 g KH2PO4 monobasic and 0.7 g K2HPO4 dibasic/liter) from bacteria grown on SM/5 plates. We determined the bacterial absorbance (OD$_{600}$) using a BioPhotometer (Eppendorf, NY) and set all suspensions to optical density (OD$_{600}$ 1.5). For experiments using lab-infected lines, we mixed the specified *Burkholderia* species at 5% and *K. pneumoniae* at 95% vol and plated *D. discoideum* spores (as indicated) with 200 µl of the bacterial mixture on SM/5 plates.

## Spot test assay

We verified infection status by spot test assay as previously described (*Brock et al., 2011*). Briefly, we transferred sorus contents from individual *D. discoideum* fruiting bodies to SM/5 agar plates using a 10 µl filter pipet tip. We incubated at 21°C for one week and checked for bacterial growth as an indication of infection.

## Fitness assay

We analyzed spore production and viability as a proxy for amoeba fitness using four sets of *D. discoideum* clones (*Table 1*). We tested three conditions: uninfected (naïve, cured naïve, cured native-*agricolaris*, and cured native-*hayleyella*), *B. agricolaris* infected (native-*agricolaris*, and naïve, native-*agricolaris*, and native-*hayleyella* first cured then infected with *B. agricolaris*) and *B. hayleyella* infected (native-*hayleyella*, and naïve, native-*agricolaris*, and native-*hayleyella* first cured then infected with *B. hayleyella*) across three temporal replicates.

To set up each assay, we plated $2 \times 10^5$ spores of each clone in each condition (with lab infected lines being plated on *Burkholderia-Klebsiella* mixtures as described) onto SM/5 agar plates in duplicate. All clones formed fruiting bodies by 3 days, so we performed data collection five days after fruiting. We used the first plate to ascertain total spore production as previously described (*Brock et al., 2011*). Briefly, spores were collected by washing plates with starvation buffer supplemented with 0.01% NP-40 alternative (Calbiochem). We counted spore dilutions on a hemocytometer using a light microscope and determined total spores according to total volume collected and dilution factor. To determine the proportion of viable spores we collected spores into starvation buffer only and determined spore density as above. We diluted suspensions to $10^4$ and spread 100 spores over ten $100 \times 15$ mm$^2$ SM/5 agar plates supplemented with 200 µl *K. pneumoniae* in starvation buffer (OD$_{600}$ 1.5). After 2 days the percentage of viable spores was determined by counting plaques formed on bacterial lawns.

## Transmission electron microscopy

We prepared amoebae by plating $2 \times 10^5$ spores (with *Klebsiella* for uninfected or native-infected and for the indicated *Burkholderia* mixture for lab-infected). We harvested log-phase vegetative cells

approximately 36 hr after plating and fruiting bodies 4 days after plating. To prepare migrating slugs, we mixed 200 µL of centrifuge-concentrated *K. pneumoniae* (absorbance, OD$_{600}$ 75) with 5 × 10$^6$ spores and plated the mixture in a straight line across a starving agar plate, which was then wrapped in aluminum foil with a small hole opposite the spore line. We incubated plates under a direct light and allowed slugs to migrate for about 80 hr before processing. We processed all stages by first adding fix solution (2% paraformaldehyde/2.5% glutaraldehyde (Polysciences Inc., Warrington, PA) in 100 mM cacodylate buffer, pH 7.2), followed by low melting agarose, over the plates to keep structures intact.

We fixed samples for 1–3 hr at room temperature then washed with cacodylate buffer and post-fixed in 1% osmium tetroxide (Polysciences Inc.) for 1 hr. We then rinsed samples extensively in dH$_2$O prior to *en bloc* staining with 1% aqueous uranyl acetate (Ted Pella Inc., Redding, CA) for 1 hr. Following several rinses in dH2O, we dehydrated samples in a graded series of ethanol and embedded them in Eponate 12 resin (Ted Pella Inc.). Sections of 95 nm were cut with a Leica Ultracut UCT ultramicrotome (Leica Microsystems Inc., Bannockburn, IL), stained with uranyl acetate and lead citrate, and viewed on a JEOL 1200 EX transmission electron microscope (JEOL USA Inc., Peabody, MA) equipped with an AMT eight megapixel digital camera (Advanced Microscopy Techniques, Woburn, MA).

## Confocal microscopy

We constructed RFP labeled versions of *B. agricolaris* (from QS70) and *B. hayleyella* (from QS11) by performing triparental mating procedures with the *E. coli* helper strain E1354 (pTNS3-asdEc) and the *E coli* donor strain E2072 with pmini-Tn7-gat-P1-RFP and confirmed glyphosate resistant RFP positive *Burkholderia* conjugants using *Burkholderia* specific PCR as previously described (*DiSalvo et al., 2015*; *Norris et al., 2009*; *Su et al., 2014*). We constructed a GFP labeled version of *K. pneumoniae* using a triparental mating strategy with the donor *E. coli* donor strain WM3064 containing pmini-Tn7-KS-GFP and the *E. coli* helper strain E1354 helper pUXBF13 as previously described (*Kikuchi and Fukatsu, 2014*; *Teal et al., 2006*). We confirmed kanamycin resistant GFP positive recipient cells by 16S rRNA gene sequencing.

Using *Burkholderia*-RFP and *Klebsiella*-GFP we infected set one hosts (using cured native hosts) as previously described. Control samples were plated with *K. pneumoniae*-GFP only. We harvested log-phase amoebae approximately 36 hr after plating by flooding plates with 5 ml SorMC and washing 3x in PBS to remove residual bacteria. We set amoebae to 1 × 10$^6$ cells/ml and placed 200 µl onto #1.5 glass coverslips for 15 min to allow them to adhere before fixing in 4% formaldehyde for 10 min. We then washed with PBS, permeabilized with 0.5% triton-X, and stained with Alexa Fluor 680 phalloidin (lifetechnologies) for 30 min before mounting in Prolong Diamond antifade mountant (lifetechnologies). We prepared spores four days after plating by collecting sori into starving buffer with 1% calcofluor white and spreading the solution on a glass bottom culture dish under a 2% agarose overlay.

We collected images using a Nikon A1Si Laser Scanning confocal microscope with a CFI Plan Apo VC Oil 1.4 NA 100X objective and Nikon Elements software or an Olympus Fluoview FV1000 confocal microscope using Plan Apo Oil 1.4NA 60X objective and Olympus software. Z-sections were taken every 0.5 microns with an average of 2 at 1024 × 1024 resolution pixels or 600 × 600 pixels. We excited RFP using the 561 laser, GFP with the 488 laser, Calcofluor-white with the 408 laser, and Alexa Fluor 680 Phalloidin with the 640 laser. We created composite images in FIJI.

## Infectivity quantification

We quantified bacterial density within individual spores by counting the number of visible *Burkholderia*-RFP cells within infected spores from confocal micrographs. We counted 15 infected spores for each condition from two individual replicates (30 spores total). We quantified the population of *Burkholderia*-RFP infected spores for the set one clones using the BD accuri C6 flow cytometer. We plated spores in duplicate as described in the confocal microscopy section. Four days after plating, we resuspended three sori from each plate into 500 µl of starving buffer with 0.01% NP-40 alternative. We ran 100 µl of each vortexed sample through the flow cytometer. We used non-fluorescent controls to establish an accurate gating between fluorescent and non-fluorescent boundaries. We measured and averaged duplicates for a total of 6 temporal replicates.

## Morphometrics

We quantified fruiting body size and shape for each clone in each condition using set one clones. We plated the clones as described under fitness assays. Five days after fruiting, we carefully cut and removed a thin strip of agar approximately 5 mm wide from the central area of an experimental plate and laid it on its side in a Petri plate. We placed dampened Kimwipes around the agar slice to prevent desiccation. We used a Leica EC3 scope with the LASD core package LAS V4.1) to collect data. Fruiting body images were taken randomly along with graticule images for calibration. We took six measurements of each fruiting body: sorus width, sorus length, stalk height and the width of the stalk at its base, midpoint and at the top just below the sorus (*Buttery et al., 2009*). Stalk height was measured from the base of stalk to the tip of the sorus. We calculated sorus volume applying the formula for the volume of a sphere using diameter, $V = 1/6 \pi d^3$. We calculated stalk volume using the formula of a cylinder, $V = \pi r^2 h$, where height (h) is the stalk height and radius (r) is half the mean of the three stalk width measurements. We measured about 80 sori and 20 stalks for each clone for each condition.

## Statistical analyses

All analyses were done in R. For fitness assays, we tested the effect of antibiotic treatment using random-slope linear mixed models (LMM) on those *D. discoideum* hosts not reinfected with *Burkholderia*. Our models included either spore numbers or proportion of viability as the response variable, host as random effect, and antibiotic treatment as a fixed effect. We similarly tested the effects of *Burkholderia* infection in the field and in the lab using random-slope LMMs on data from hosts cured with antibiotics. Our models included spore numbers or proportion viability as response variable, host as random effect, lab or field-infection status and *Burkholderia* type as fixed effects, as well as an interaction between field-infection status and *Burkholderia* infection. For all LMMs, we fitted models and assessed model fit with likelihood ratio tests executed with the lme4 package (*Bates et al., 2015*) in the R environment (v. 3.3.3, R Core Team 2017). We tested the significance of fixed effects with Wald tests using the *t* distribution, which we executed with the packager lmerTest (*Kuznetsova et al., 2017*). These tests use (*Satterthwaite, 1946*) approximation for denominator degrees of freedom to calculate *p*-values. Finally, for all *post hoc* multiple comparisons, we performed pairwise contrasts of least-square means with a multivariate *t* distribution adjustment as implemented with the package lsmeans (*Lenth, 2016*).

For morphometric analyses, we also tested the effect of antibiotic treatment on those hosts not re-infected with *Burkholderia*. We used a 2-way analysis of variance (ANOVA) with one of our four morphological measurements as the response variable and both *Burkholderia* colonization and antibiotic treatment as fixed effects. Similarly, we tested the effects of *Burkholderia* colonization from the field and *Burkholderia* infection in the lab on the amoeba hosts with 2-way ANOVAs on amoebae cured with antibiotics. Again, one of the four morphological measurements was the response variable with field-colonization status, *Burkholderia* infection, and an interaction between them as fixed effects. For all ANOVAs, when appropriate, we performed *post hoc* Tukey HSD tests for multiple comparisons. Sorus width data were square-root transformed as $\sqrt{x+2}$ and sorus volume data were $\log_e$-transformed to meet test assumptions of normally distributed residuals.

We analyzed *Burkholderia*-RFP infectivity (population prevalence and intracellular density) with 2-way ANOVAs, followed by a Tukey HSD for multiple comparisons. We treated *Burkholderia* species and *D. discoideum* identity as fixed effects. All source data is available at https://doi.org/10.7936/wgnk-2c37.

## Acknowledgements

Many thanks to the Strassmann/Queller laboratory group for useful discussion and Mountain Lake Biological Station where we collected the samples. Thanks to Kyle Skottke, Hassan Salem, and Tyler Larsen for helpful comments on the manuscript. We also thank the reviewers of this manuscript for their thorough analysis and insightful suggestions. All authors read and approved the final manuscript.

This material is based upon work supported by the National Science Foundation under grant nos. NSF DEB1146375, NSF IOS 1256416, NSF IOS 1656756, the Life Sciences Research Foundation and Simons Foundation (to LS), and the John Templeton Foundation grant no. 43667.

## Additional information

### Funding

| Funder | Grant reference number | Author |
|---|---|---|
| National Science Foundation | DEB1146375 | David C Queller<br>Joan E Strassmann |
| Life Sciences Research Foundation | | Longfei Shu |
| John Templeton Foundation | 43667 | David C Queller<br>Joan E Strassmann |
| National Science Foundation | IOS1256416 | David C Queller |
| National Science Foundation | IOS1656756 | David C Queller |
| Simons Foundation | | Longfei Shu |

The funders had no role in study design, data collection and interpretation, or the decision to submit the work for publication.

### Author contributions

Longfei Shu, Conceptualization, Investigation, Methodology; Debra A Brock, Conceptualization, Validation, Investigation, Methodology; Katherine S Geist, Data curation, Formal analysis, Validation; Jacob W Miller, Investigation; David C Queller, Conceptualization, Formal analysis, Supervision, Funding acquisition, Project administration, Writing—review and editing; Joan E Strassmann, Conceptualization, Resources, Data curation, Supervision, Funding acquisition, Methodology, Project administration, Writing—review and editing; Susanne DiSalvo, Conceptualization, Data curation, Investigation, Visualization, Methodology, Writing—original draft, Writing—review and editing

### Author ORCIDs

Longfei Shu http://orcid.org/0000-0001-9683-906X
Debra A Brock http://orcid.org/0000-0002-4349-5854
Susanne DiSalvo http://orcid.org/0000-0002-4001-4672

### Decision letter and Author response

Decision letter https://doi.org/10.7554/eLife.42660.022
Author response https://doi.org/10.7554/eLife.42660.023

## Additional files

### Supplementary files

• Supplementary file 1. Statistical results of three fitness measures assayed for field-collected amoeba clones and after curing with antibiotics. The three fitness measures were percent of spores that were viable, the total number of spores produced by a clone, and total viable spores. Total viable spores is the product of the other two measures. For each pairwise contrast, the essential difference in treatments is in boldface, and a treatment that is significantly higher than the other is marked with an asterisk and printed in red. Each of the fitness measures was analyzed with a set of Generalized Linear Mixed Models (GLMMs). This table gives the $p$-values for each question asked about main or interaction effects and the *post hoc* pairwise comparisons made, as relevant. Details about the statistical tests used can be found in the main text.
DOI: https://doi.org/10.7554/eLife.42660.014

• Supplementary file 2. Statistical results of three fitness measures assayed for antibiotic-cured amoeba clones after experimental addition of *Burkholderia*. he three fitness measures were again percent of spores that were viable, the total number of spores produced by a clone, and total viable spores. Total viable spores is the product of the other two measures. For each pairwise contrast, the essential difference in treatments is in boldface, and a treatment that is significantly higher than the other is marked with an asterisk and printed in red. Each of the fitness measures was analyzed with a set of Generalized Linear Mixed Models (GLMMs). This table gives the *p*-values for each question asked about main or interaction effects and the *post hoc* pairwise comparisons made, as relevant. Details about the statistical tests used can be found in the main text.

DOI: https://doi.org/10.7554/eLife.42660.015

• Supplementary file 3. Statistical results for stalk morphology. Each of the stalk measures was analyzed with a set of Generalized Linear Mixed Models (GLMMs). This table gives the *p*-values for each question asked about main or interaction effects and the *post hoc* pairwise comparisons made, as relevant. For each pairwise contrast, the essential difference in treatments is in boldface, and a treatment that is significantly higher than the other is marked with an asterisk and printed in red. Details about the statistical tests used can be found in the main text. One clone of each native type was tested: QS9 naïve; QS70 *ag*-infected; QS11 *ha*-infected.

DOI: https://doi.org/10.7554/eLife.42660.016

• Supplementary file 4. Statistical results for sorus morphology. Each of the spore measures was analyzed with a set of Generalized Linear Mixed Models (GLMMs). This table gives the *p*-values for each question asked about main or interaction effects and the *post hoc* pairwise comparisons made, as relevant. For each pairwise contrast, the essential difference in treatments is in boldface, and a treatment that is significantly higher than the other is marked with an asterisk and printed in red. Details about the statistical tests used can be found in the main text. One clone of each native type was tested: QS9 naïve; QS70 *ag*-infected; QS11 *ha*-infected.

DOI: https://doi.org/10.7554/eLife.42660.017

• Transparent reporting form

DOI: https://doi.org/10.7554/eLife.42660.018

## Data availability

All raw data has been archived in the Washington University Library: https://doi.org/10.7936/wgnk-2c37

The following dataset was generated:

| Author(s) | Year | Dataset title | Dataset URL | Database and Identifier |
|---|---|---|---|---|
| Strassmann J, DiSalvo S, Queller DC, Brock DA | 2018 | Raw data from Symbiont location, host fitness, and possible coadaptation in a symbiosis between social amoebae and bacteria | https://openscholarship.wustl.edu/data/11/ | Washington University Open Scholarship, 10.7936/wgnk-2c37 |

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
