## [Decision Letter]

Thank you for submitting your article "A complex symbiosis involving within species variation in the response of *Dictyostelium* amoebae to *Burkholderia* bacteria" for consideration by *eLife*. Your article has been reviewed by three peer reviewers, including Antonio Ruzzini as the Reviewing Editor and Reviewer #1, and the evaluation has been overseen by Ian Baldwin as the Senior Editor. The following individuals involved in review of your submission have agreed to reveal their identity: Adam Kuspa (Reviewer #2); Eric Harvill (Reviewer #3).

The reviewers have discussed the reviews with one another and the Reviewing Editor has drafted this decision to help you prepare a revised submission.

Summary:

Through a simple but elegant experimental design to measure variation in *Dictyostelium's* responses to different *Burkholderia*, rendered interpretable with a sophisticated statistical treatment, the authors arrive at the important conclusion that amoebal physiology can be evolutionarily shaped by its naturally occurring symbionts. The authors describe biological phenomena that result from *Dictyostelium discoideum* colonization by symbiotic *Burkholderia* spp. (*B. agricolaris* and *B. hayleyella*) using a laboratory model of infection. The colonization of both field-collected naïve and infected protists was systematically assessed to define the relationships between pairs of micro-organisms. Antibiotics were used to cure the amoebae from their respective co-isolated bacterial symbionts, and the naturally symbiont-free (naïve) and cured D*. discoideum* were subsequently infected with one of three *Burkholderia* spp. A comparison of host reproductive fitness, fruiting boding morphology, symbiont localization and abundance during uni- and multicellular life stages, and the subcellular effects of bacteria on four distinct hosts is summarized. The authors describe species-specific correlations between host-exposure and fitness upon curing and re-infection with an individual bacterium. A number of the effects of colonization are visualized using confocal and transmission electron microscopy. Overall, the results show a trend consistent with co-adaptation among ecological (co-isolated) pairs. In the aggregate, co-isolated protist-bacterium pairs appear healthier under laboratory conditions than the mixed pairs generated by substituting one of the co-isolated species for an ecologically disparate partner. Differences between the two *Burkholderia* spp. are also highlighted: the *B. agricolaris* symbionts only moderately decreased fitness whereas infection with *B. hayleyella* more drastically impacted the amoebae, particularly naïve and previously *B. agricolaris*-harboring clones.

It is clear that this report will set the stage for a number of future investigations into the protist-bacterium symbiosis. Although *D. discoideum* has served as a model laboratory organism for decades, including as a host to investigate virulence among bacterial pathogen, this report helps to establish the system as an invaluable experimental model to study naturally occurring symbioses.

Essential revisions:

There were several questions related to the origin and description of the organisms used in the study (see comments 1-5).

1) This work provides greater insight into the ability of *D. discoideum*-associated *Burkholderia* to colonize hosts, including those that were devoid of bacteria within their sori at the time of harvest. In the past, these amoeba were named non-farmers, used to contrast the discovery of amoebae capable of carrying both food and non-food (symbiotic) bacteria named farmers. Here, the authors refer to the farmers as natively infected. The non-farmers are referred to as naïve. Readers that are familiar with the previous nomenclature would benefit from including these distinctions. One concern is that the manuscript reads as if the authors harvested *D. discoideum* explicitly for this work. The clone identifiers, however, match those present in the published literature (e.g. the farmer/native-ha QS11). It would be beneficial to clarify the differences in nomenclature and the origin of the organisms within the body of the text.

2) The relatively clear separation between the three groups of amoebae suggests that they may belong to (at least) three different evolutionary lineages or even subspecies. Can the authors please provide typing data to show that there is no difference or to differentiate between the *D. discoideum* groups?

3) Similarly, do all symbiotic bacteria belong to a single sequence type (ST) per species or do they represent independent bacterial clones?

4) Table 1. All but two samples were collected at the Mt Lake Biological Station in Virginia. What was the rationale for sampling two of the *D. discoideum* with *B. agricolaris* from Texas and North Carolina? Were all samples collected in the same year?

5) *B. agricolaris* strains were isolated from clones QS70, QS159, QS161 and NC21. Can the authors please clarify which of these strains was used for the reinfections? The same for *B. hayleyella*?

6) The experimental differences between the 'field-harvested' and 're-infected' samples are somewhat unclear. The description of the laboratory (re)infection is clear but the manipulation of the 'field harvested' spores is not. Are these grown without the addition of additional food bacteria? If so, then is it possible to quantify the *Burkholderia*:Food ratio present in these inoculations? This information would help to provide greater context to the detrimental effects of the symbionts on the hosts and the potential limitations of the current experimental system.

7) I see no glaring issues with the current manuscript but I have one suggestion that may improve our understanding. I would not insist that the experiment be added to the revised manuscript.

Figure 7B, C is about a higher percentage of spores being infected, but it also looks like there are more bacteria per spore when the amoebae are re-infected with their "natural" symbiont compared to the other *Burkholderia* strain. One cannot be sure based on the one representative panel shown (not a criticism), but this could be determined by flow cytometry as a peaks of spores containing more or less bacteria per spore. It would be important because it would provide another piece of evidence that co-evolution or tuning of the symbiosis is occurring. Also, the fitness detriment could be made worse by more bacteria per spore (perhaps this is what was meant by the authors use of the phrase "bacterial density" (end of subsection “*Burkholderia* location inside *D. discoideum*”, this is ambiguous because it could refer to density within the sorus or within each spore).

Nice-to-have revisions:

8) Some estimations about the bacteria visible intracellular in TEM could be readily verified via gold-labeled antibodies.

9) Additional experiments that further validate the potential of the system would strengthen the manuscript. The serial passage of newly established (post-curing) bacterium-protist pairs could be used to compare the initial and re-infection phenotypes to those of the natively infected individuals. An investigation of sexual reproduction and symbiont status would also interests readers, however, such an undertaking falls outside of the scope of this submission.

---

## [Author Response]

Essential revisions:There were several questions related to the origin and description of the organisms used in the study (see comments 1-5).1) This work provides greater insight into the ability of D. discoideum-associated Burkholderia to colonize hosts, including those that were devoid of bacteria within their sori at the time of harvest. In the past, these amoeba were named non-farmers, used to contrast the discovery of amoebae capable of carrying both food and non-food (symbiotic) bacteria named farmers. Here, the authors refer to the farmers as natively infected. The non-farmers are referred to as naïve. Readers that are familiar with the previous nomenclature would benefit from including these distinctions. One concern is that the manuscript reads as if the authors harvested D. discoideum explicitly for this work. The clone identifiers, however, match those present in the published literature (e.g. the farmer/native-ha QS11). It would be beneficial to clarify the differences in nomenclature and the origin of the organisms within the body of the text.

We agree that the terminology is overall a bit confusing. We elected to use “naïve” and “native” as opposed to farmers and non-farmers in light of our more recent work demonstrating that *Burkholderia* symbionts endow farming traits to non-farmers. This complicates the use of the farming terminology because it can either be interpreted in reference to the original field infected state of an amoebae isolate, or in reference to any isolate subsequently infected with *Burkholderia* (whether infected in the field or in lab). Field-infected clones have had an unknown number of generations to deal with their bacterial indwellers. We agree that adding reference to previous nomenclature benefits the overall clarity of the paper and provides context for readers familiar with the field. To this aim we have added the following sentence to the Introduction:

“Infection induces secondary carriage of edible bacteria (*Burkholderia* is typically inedible), which can increase host fitness in food scarce conditions when the carried food bacteria reseed new environments with a food source (Brock et al., 2011; DiSalvo et al., 2015). For this reason, field-infected host amoebae are also called “farmers” and uninfected host amoebae are called “non-farmers.”

We also see how our description of the amoebae isolates could lead a reader to infer that we harvested them explicitly for this work. This was an unintentional consequence of our attempt at conciseness.

To clarify this, we added the following to the Introduction:

"Here we probe the interaction between host and symbiont genotypes (with regards to their association history) with infection outcomes using a set of previously field-collected and characterized amoebae isolates and their associated *Burkholderia* symbionts (Brock et al., 2011; DiSalvo et al., 2015). "

And modified a sentence within the Results section to read:

"We used 12 natural isolates of *Dictyostelium*, each from three original conditions, uninfected with *Burkholderia*, infected with *B. agricolaris*, or infected with *B. hayleyella* (Table 1)."

2) The relatively clear separation between the three groups of amoebae suggests that they may belong to (at least) three different evolutionary lineages or even subspecies. Can the authors please provide typing data to show that there is no difference or to differentiate between the D. discoideum groups?

We have previously sequenced several genes for phylogenetic analysis of the four naïve and the four native-*hayleyella* amoebae isolates used in this study (Brock et al., 2011) and shown them all to be *D. discoideum* and to not represent individual groups within the species. Furthermore, we can place strains carrying either *B. hayleyella* or *B. agricolaris* or neither in the phylogeny from Douglas et al. 2011. We were able to put 4 *B. agricolaris* carriers, 8 B*. hayleyella* carriers, and the 4 non carriers used in this study in that phylogeny. Clearly there is no phylogenetic differentiation defined by host type.

We have added the following sentences to the Discussion section of this paper:

“Previous genetic analysis of a collection of *D. discoideum* isolates revealed no phylogenetic differentiation between naive, native-*agricolaris*, and native-*hayleyella* host types (Douglas et al., 2011). We envision that diverse amoebae isolates get infected with *Burkholderia* in nature, and then adaptations that provide these hosts with increased resilience are selected for over time.”.

3) Similarly, do all symbiotic bacteria belong to a single sequence type (ST) per species or do they represent independent bacterial clones?

All *Burkholderia* strains used for this study have been multi-locus sequence typed (Haselkorn et al., 2018). From these sequencing data, *B. agricolaris* strains Ba70 and Ba161 are the same haplotype and Ba159 and BaNC21 are unique haplotypes. However, none of the *B. hayleyella* strains we have in our collection vary in any of the multi-locus genes so far sequenced. Thus, the four *B. hayleyella* strains used in this study all represent the same haplotype. However, we anticipate that there may be sequence differences between these strains as they were isolated from different amoebae hosts but do not have full genome sequences from all these strains to say definitively.

We have added the following sentence to the Materials and methods section to clarify this:

"According to multi-locus sequence typing all *B. hayleyella* strains belong to the same haplotype, *B. agricolaris* from QS70 and QS161 are the same haplotype, and *B. agricolaris* from NC21 and QS159 are unique haplotypes (Haselkorn et al., 2018)."

4) Table 1. All but two samples were collected at the Mt Lake Biological Station in Virginia. What was the rationale for sampling two of the D. discoideum with B. agricolaris from Texas and North Carolina? Were all samples collected in the same year?

As we did not explicitly collect new clones for this study, we did not consider sampling location and date when selecting our clone set. Instead, host-symbiont pairs from this set were mainly used because they had been used in our previous analyses, and we wanted to add to our expanding knowledge of these genotypes. Most clones were collected in 2000, with QS70 collected in 2004, and NC21 collected in October 1988.

5) B. agricolaris strains were isolated from clones QS70, QS159, QS161 and NC21. Can the authors please clarify which of these strains was used for the reinfections? The same for B. hayleyella?

We used a specific reinfection strategy for the fitness assays that was somewhat challenging to describe concisely within the text of the manuscript. This reinfection strategy corresponds to the clone “sets” listed in table one. To better clarify these reinfections for the fitness assays (wherein multiple clone types were used) we added the following:

"To test the effects of *Burkholderia* on all types of field-collected hosts, we grouped host types into four sets, with each set representing a unique naïve, native-*agricolaris*, and native-*hayleyella* host (Table 1). […] From these groups, we compared viable spore production of cured *D. discoideum* hosts versus those same hosts artificially infected with *B. agricolaris* or *B. hayleyella* (Figure 2B, C, D)."

"We then asked whether *D. discoideum* hosts are adapted to the *Burkholderia* species they carried in the field (using the same set strategy described above)."

What we are indicating here is that clone-*Burkholderia* pairs were swapped within sets (according to Table 1). For example: For set one clones, QS9 was infected with Bh11 or Ba70, QS11 was reinfected with Bh11 or Ba70, and QS70 was reinfected with Bh11 or Ba70. For set two clones, QS18 was infected with Ba159 or Bh23, QS159 was reinfected with Ba159 or Bh23, and QS23 was reinfected with Ba159 or Bh23 (and so on). We hope this is clarified by our above additions.

In the case of electron and confocal microscopy, only the set one clones were used (QS9, QS11, and QS70) along with their corresponding symbionts (Bh11 and Ba70).

6) The experimental differences between the 'field-harvested' and 're-infected' samples are somewhat unclear. The description of the laboratory (re)infection is clear but the manipulation of the 'field harvested' spores is not. Are these grown without the addition of additional food bacteria? If so, then is it possible to quantify the Burkholderia:Food ratio present in these inoculations? This information would help to provide greater context to the detrimental effects of the symbionts on the hosts and the potential limitations of the current experimental system.

We can see how the use of "field harvested" could be interpreted to mean that spores came directly from the field. In this case, we are using the term to indicate that apart from maintaining these clones normally in the laboratory environment (by growing on nutrient media with bacterial food, *Klebsiella pneumoniae*) we have not intentionally manipulated them by curing them of bacterial associates or exposing them to any non-food bacteria. Thus, in all cases, *Dictyostelium* clones come from frozen spore stocks and are pre-grown with *K. pneumoniae* before each experimental assay. For all assays, all clone types are either grown only with *K. pneumoniae* food (for field harvested or cured clones) or are grown with *Burkholderia* supplemented *K. pneumoniae* mixtures (to infect naive or reinfect cured clones). In each inoculation, we set all bacterial suspensions to an OD600 of 1.5. For *Burkholderia*:Food inoculations, we used 5% *Burkholderia* with 95% *K. pneumoniae* by volume (as described in the Materials and methods section). In the case of native field infected clones, we do not have a quantification for the *Burkholderia*:food ratio, as each clone is carrying in its already pre-established dose of *Burkholderia*. Because of the possibility that native hosts could be inoculated with a different dose of *Burkholderia* than we are providing for artificial infections, in addition to making comparisons between naïve and native hosts in their field infected conditions we analyze cured and reinfected hosts. In the former condition, there is the possibility that natural infection densities play a role in distinct host outcomes, while in the later condition, symbiont densities are controlled to reveal differences in host outcomes that are products of factors other than symbiont exposure loads.

In order to clarify this point, we added the following to the Results section:

"The word “host” always means *D. discoideum*, and sometimes refers to potential hosts not actually infected with *Burkholderia*. Prior to each treatment, all host types were developed on nutrient media supplemented with bacterial food (*K. pneumoniae*). Amoebae, spores, cells, slugs, fruiting bodies, stalks, and sori refer only to *D. discoideum*."

7) I see no glaring issues with the current manuscript but I have one suggestion that may improve our understanding. I would not insist that the experiment be added to the revised manuscript.Figure 7B, C is about a higher percentage of spores being infected, but it also looks like there are more bacteria per spore when the amoebae are re-infected with their "natural" symbiont compared to the other Burkholderia strain. One cannot be sure based on the one representative panel shown (not a criticism), but this could be determined by flow cytometry as a peaks of spores containing more or less bacteria per spore. It would be important because it would provide another piece of evidence that co-evolution or tuning of the symbiosis is occurring. Also, the fitness detriment could be made worse by more bacteria per spore (perhaps this is what was meant by the authors use of the phrase "bacterial density" (end of subsection “Burkholderia location inside D. discoideum”, this is ambiguous because it could refer to density within the sorus or within each spore).

We modified some of the writing to clarify infection density vs. infection prevalence in spore populations. We agree that a thorough analysis exploring the number of bacteria per spore in different hosts would help uncover potential differences in host control over symbiont amplification (a speculated mechanistic possibility for why *B. hayleyella* hosts may be less damaged by their symbiont). Our flow cytometry approach did not show an obvious shift in fluorescence intensity among different strains, so instead we tried to address this as best possible by counting visible bacterial cells per spore in the confocal images generated for this study. We have added these data to Figure 7 (Figure 7E) and described in several areas of the manuscript as illustrated below.

Results section:

"In line with the visualized differences between the abundance of the two species of *Burkholderia* in amoebae, we find significantly more naïve spores infected with RFP labeled *B. hayleyella* (mean= 88.8%) than similarly labeled *B. agricolaris* (mean= 35.3%) (F_2,4_ = 191.33, P < 0.001) (Figure 7D). […] However, B*. agricolaris* displays a significantly higher intracellular density (mean=10.55) within individual infected spores than *B. hayleyella* (mean=7.45) (F_1,177_=23.845, p<0.001), while host background does not play a significant role in *Burkholderia* intracellular density (F_2,177_=2.723, p=0.068) (Figure 7E). "

Materials and methods section:

"We quantified bacterial density within individual spores by counting the number of visible *Burkholderia*-RFP cells within infected spores from confocal micrographs. We counted 15 infected spores for each condition from two individual replicates (30 spores total)."

Discussion section:

"The percent of spore cells infected in the population post symbiont exposure are not significantly different between a native-*hayleyella* host and naïve hosts, which does not support the idea that infection events are inhibited. […] However, our TEM analysis qualitatively points to the idea that after infection, intracellular replication rates may differ between naïve and native-*hayleyella* hosts."

Although we were unable to detect a difference in bacterial densities between naive and native hosts for *B. hayleyella* infections, it is still possible that *Burkholderia* cells are amplifying more in naive hosts, but are perhaps inducing lysis (or spore development inhibition) in those more densely infected cells such that we don't catch this difference at the spore stage. Our confocal and TEM analysis of other developmental stages qualitatively suggest this, but a more sensitive and higher throughput strategy (possibly qPCR assays) may be necessary to properly investigate this possibility.

Nice-to-have revisions:8) Some estimations about the bacteria visible intracellular in TEM could be readily verified via gold-labeled antibodies.

This certainly would be a nice addition but is difficult to accomplish at this point. We felt that the intracellular bacterial cells in TEM were sufficiently identifiable morphologically. To bolster our interpretations of bacterial cells from TEM, we used confocal microscopy for some of the same key stages using *Burkholderia* labeled with red fluorescent protein (allowing for *Burkholderia* identification in specific). Both of these assays aligned with one another, making us more confident with our TEM interpretation.

9) Additional experiments that further validate the potential of the system would strengthen the manuscript. The serial passage of newly established (post-curing) bacterium-protist pairs could be used to compare the initial and re-infection phenotypes to those of the natively infected individuals. An investigation of sexual reproduction and symbiont status would also interests readers, however, such an undertaking falls outside of the scope of this submission.

We have previously shown that *Dictyostelium* specific *Burkholderia* is passaged effectively over the *Dictyostelium* social cycle in new and native hosts (Haselkorn et al., 2018). Serial passaging all clone types after reinfection to analyze phenotypes would be interesting, but would require a time investment beyond our current ability. Investigations into symbiont status post sexual reproduction would also be interesting, however there are no techniques that have been developed (despite efforts) to consistently induce sexual reproduction and recombination in the lab setting.